# Domain Adaptation under Open Set Label Shift

**Saurabh Garg, Sivaraman Balakrishnan, Zachary C. Lipton**
Machine Learning Department,
Department of Statistics and Data Science,
Carnegie Mellon University
{sgarg2,sbalakri,zlipton}@andrew.cmu.edu

## Abstract

We introduce the problem of domain adaptation under Open Set Label Shift (OSLS) where the label distribution can change arbitrarily and a new class may arrive during deployment, but the class-conditional distributions $p(x|y)$ are domain-invariant. OSLS subsumes domain adaptation under label shift and Positive-Unlabeled (PU) learning. The learner's goals here are two-fold: (a) estimate the target label distribution, including the novel class; and (b) learn a target classifier. First, we establish necessary and sufficient conditions for identifying these quantities. Second, motivated by advances in label shift and PU learning, we propose practical methods for both tasks that leverage black-box predictors. Unlike typical open set domain adaptation problems, which tend to be ill-posed and amenable only to heuristics, OSLS offers a well-posed problem amenable to more principled machinery. Experiments across numerous semi-synthetic benchmarks on vision, language, and medical datasets demonstrate that our methods consistently outperform open set domain adaptation baselines, achieving 10–25% improvements in target domain accuracy. Finally, we analyze the proposed methods, establishing finite-sample convergence to the true label marginal and convergence to optimal classifier for linear models in a Gaussian setup[1].

## 1   Introduction

Suppose that we wished to deploy a machine learning system to recognize diagnoses based on their clinical manifestations. If the distribution of data were static over time, then we could rely on the standard machinery of statistical prediction. However, disease prevalences are constantly changing, violating the assumption of independent and identically distributed (iid) data. In such scenarios, we might reasonably apply the *label shift* assumption, where prevalences can change but clinical manifestations cannot. When only the relative proportion of previously seen diseases can change, principled methods can detect and correcting for label shift on the fly [56, 78, 45, 4, 1, 27]. But what if a new disease, like COVID-19, were to arrive suddenly?

Traditional label shift adaptation techniques break when faced with a previously unseen class. A distinct literature on Open Set Domain Adaptation (OSDA) seeks to handle such cases [51, 5, 14, 70, 43, 73, 58, 59, 25]). Given access to labeled *source* data and unlabeled *target* data, the goal in OSDA is to adapt classifiers in general settings where previous classes can shift in prevalence (and even appearance), and novel classes separated out from those previously seen can appear. Most work on OSDA is driven by the creation of and progress on benchmark datasets (e.g., DomainNet, OfficeHome). Existing OSDA methods are heuristic in nature, addressing settings where the right answers seem intuitive but are not identified mathematically. However, absent assumptions on: (i) the nature of distribution shift among source classes and (ii) the relation between source classes and novel class, standard impossibility results for domain adaptation condemn us to guesswork [8].

---

[1]Code is available at `https://github.com/acmi-lab/Open-Set-Label-Shift`.

36th Conference on Neural Information Processing Systems (NeurIPS 2022).

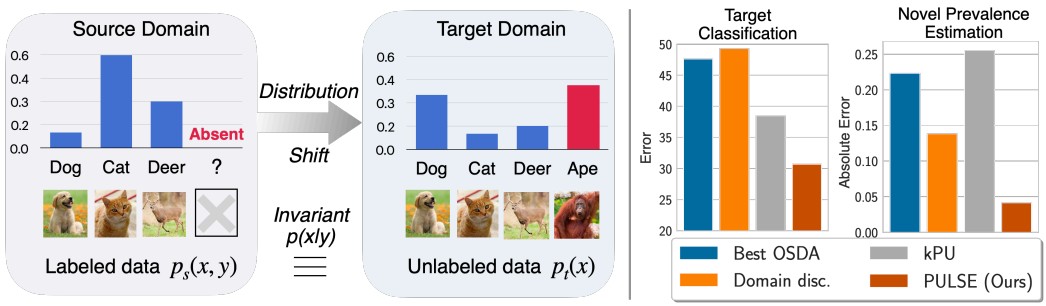

Figure 1: **Left:** *Domain Adaptation under OSLS*. An instantiation of OSDA that applies label shift assumption but allows for a new class to show up in target domain. **Right:** *Aggregated results across seven semi-synthetic benchmark datasets*. For both target classification and novel class prevalence estimation, PULSE significantly outperforms other methods (lower error is better). For brevity, we only include result for the best OSDA method. For detailed comparison, refer Sec. 7.

In this work, we introduce domain adaptation under Open Set Label Shift (OSLS), a coherent instantiation of OSDA that applies the label shift assumption but allows for a new class to show up in the target distribution. Formally, the label distribution may shift between source and target $p_s(y) \neq p_t(y)$, but the class-conditional distributions among previously seen classes may not (i.e., $\forall y \in \{1, 2, \ldots, k\}, p_s(x|y) = p_t(x|y)$). Moreover, a new class $y = k+1$ may arrive in the target period. Notably, OSLS subsumes label shift [56, 67, 45] (when $p_t(y = k+1) = 0$) and learning from Positive and Unlabeled (PU) data [20, 42, 24] (when $k = 1$). As with label shift and PU learning, our goals are two-fold. Here, we must (i) estimate the target label distribution $p_t(y)$ (including the novel class prevalence); (ii) train a $(k+1)$-way target-domain classifier.

First, we characterize when the parameters of interest are identified (Sec. 4). Namely, we define a (necessary) *weak positivity* condition, which states that there exists a subset of each label's support that has zero probability mass under the novel class and that the submatrix of $p(x|y)$ consisting only of rows in that subset is full rank. Moreover, we prove that weak positivity alone is not sufficient. We introduce two sufficient conditions: *strong positivity* and *separability*, either of which (independently) ensures identifiability.

Focusing on cases with strong positivity, we show that OSLS reduces to $k$ PU learning problems (Sec. 5). However, we demonstrate that straightforward applications of this idea fail because (i) bias accumulates across the $k$ mixture proportion estimates leading to grossly underestimating the novel class's prevalence; and (ii) naive combinations of the $k$ PU classifiers are biased and inaccurate.

Thus motivated, we propose the PULSE framework, which combines methods from Positive and Unlabeled learning and Label Shift Estimation, yielding two-stage techniques for both label marginal estimation and classification (Sec. 6). Our methods build on recent advances in label shift [45, 4, 1, 27] and PU learning [38, 35, 29], that leverage appropriately chosen black-box predictors to avoid the curse of dimensionality. PULSE first estimates the label shift among previously seen classes, and then re-samples the source data to formulate a single PU learning problem between (reweighted) source and target data to estimate fraction of novel class and to learn the target classifier. In particular, our procedure builds on the BBE and CVIR techniques proposed in Garg et al. [29]. PULSE is simple to implement and compatible with arbitrary hypothesis classes (including deep networks).

We conduct extensive semi-synthetic experiments adapting seven benchmark datasets spanning vision (CIFAR10, CIFAR100, Entity30), natural language (Newsgroups-20), biology (Tabula Muris), and medicine (DermNet, BreakHis) (Sec. 7). Across numerous data modalities, draws of the label distributions, and model architectures, PULSE consistently outperforms generic OSDA methods, improving by 10–25% in accuracy on target domain. Moreover, PULSE outperforms methods that naively solve $k$ PU problems on both label distribution estimation and classification.

Finally, we analyze our framework (Sec. 8). First, we extend Garg et al. [29]'s analysis of BBE to derive finite-sample error bounds for our estimates of the label marginal. Next, we develop new analyses of the CVIR objective [29] that PULSE relies in the classification stage. Focusing on a Gaussian setup and linear models optimized by gradient descent, we prove that CVIR converges to

a true positive versus negative classifier in population. Addressing the overparameterized setting where parameters exceed dataset size, we conduct an empirical study that helps to elucidate why, on separable data, CVIR outperforms other consistent objectives, including nnPU [38] and uPU [21].

## 2  Related Work

**(Closed Set) Domain Adaptation (DA)**   Under DA, the goal is to adapt a predictor from a source distribution with labeled data to a target distribution from which we observe only unlabeled examples. DA is classically explored under two distribution shift scenarios [67]: (i) Covariate shift [78, 74, 19, 18, 32] where $p(y|x)$ remains invariant among source and target; and (ii) Label shift [56, 45, 4, 1, 27, 77] where $p(x|y)$ is shared across source and target. In these settings most theoretical analysis requires that the target distribution's support is a subset of the source support [8]. However, recent empirically work in DA [48, 49, 68, 69, 80, 79, 26, 64] focuses on settings motivated by benchmark datasets (e.g., WILDS [57, 39], Office-31 [55] OfficeHome [71], DomainNet [52]) where such overlap assumptions are violated. Instead, they rely on some intuitive notion of semantic equivalence across domains. These problems are not well-specified and in practice, despite careful hyperparameter tuning, these methods often do not improve over standard empirical risk minimization on source data alone for practical, and importantly, previously unseen datasets [57].

**Open Set Domain Adaptation (OSDA)**   OSDA [51, 9, 62] extends DA to settings where along with distribution shift among previously seen classes, we may observe a novel class in the target data. This setting is also known as *universal domain adaptation* [73, 59]. Rather than making precise assumptions about the nature of shift between source and target as in OSLS, the OSDA literature is primarily governed by semi-synthetic problems on benchmark DA datasets (e.g. DomainNet, Office-31 and OfficeHome). Numerous OSDA methods have been proposed [5, 14, 70, 43, 73, 58, 59, 25, 11]. At a high level, most OSDA methods perform two steps: (i) align source and target representation for previously seen classes; and (ii) train a discrimination to reject novel class from previously seen classes. The second step typically uses novelty detection heuristics to identify novel samples.

**PU Learning**   Positive and Unlabeled (PU) learning is the base case of OSLS. Here, we observe labeled data a single source class and unlabeled target data contains data from both the novel class and the source class. In PU learning, our goals are: (i) Mixture Proportion Estimation (MPE), i.e., determining the fraction of previously seen class in target ; and (ii) PU classification, i.e., learning to discriminate between the novel and the positive (source) class. Several classical methods were proposed for both MPE [24, 23, 63, 36, 6, 7] and classification [24, 22, 21]. However, classical MPE methods do not scale to high-dimensional settings [53]. More recent methods alleviate these issues by operating in classifier output space [29, 35]. For classification, traditional methods fail when deployed with models classes with high capacity due to their capacity of fitting random labels [75]. Recent methods [29, 38, 16], avoid over-fitting by employing regularization or self-training techniques.

**Other related work**   A separate line of work looks at the problem of Out-Of-Distribution (OOD) detection [34, 31, 41, 37, 50, 76]. Here, the goal is to identify novel examples, i.e., samples that lie out of the support of training distribution. The main different between OOD detection and OSDA is that in OOD detection we do not have access to unlabeled data containing a novel class. Recently, Cao et al. [13] proposed open-world semi-supervised learning, where the task is to not only identify novel classes in target but also to separate out different novel classes in an unsupervised manner.

Our work takes a step back from the hopelessly general OSDA setup, introducing OSLS, a well-posed OSDA setting where the sought-after parameters can be identified.

## 3  Open Set Label Shift

**Notation**   For a vector $v \in \mathbb{R}^d$, we use $v_j$ to denote its $j^{\text{th}}$ entry, and for an event $E$, we let $\mathbb{I}[E]$ denote the binary indicator of the event. By $|A|$, we denote the cardinality of set $A$.

Let $\mathcal{X} \in \mathbb{R}^d$ be the input space and $\mathcal{Y} = \{1, 2, \ldots, k+1\}$ be the output space for multiclass classification. Let $\mathrm{P}_s$ and $\mathrm{P}_t$ be the source and target distributions and let $p_s$ and $p_t$ denote the corresponding probability density (or mass) functions. By $\mathbb{E}_s$ and $\mathbb{E}_t$, we denote expectations over the source and target distributions. We assume that we are given a loss function $\ell : \Delta^k \times \mathcal{Y} \to \mathbb{R}$, such that $\ell(z, y)$ is the loss incurred by predicting $z$ when the true label is $y$. Unless specified otherwise, we assume that $\ell$

is the cross entropy loss. As in standard unsupervised domain adaptation, we are given independently and identically distributed (iid) samples from labeled source data $\{(x_1, y_1), (x_2, y_2), \ldots, (x_n, y_n)\} \sim \mathrm{P}_s^n$ and iid samples from unlabeled target data $\{x_{n+1}, x_{n+2}, \ldots, x_{n+m}\} \sim \mathrm{P}_t^m$.

Before formally introducing OSLS, we describe label shift and PU learning settings. Under label shift, we observe data from $k$ classes in both source and target where the conditional distribution remain invariant (i.e., $p_s(x|y) = p_t(x|y)$ for all classes $y \in [1, k]$) but the target label marginal may change (i.e., $p_t(y) \neq p_s(y)$). Additionally, for all classes in source have a non-zero support , i.e., for all $y \in [1, k]$, $p_s(y) \geqslant c$, where $c > 0$. Under PU learning, we possess labeled source data from a positive class and unlabeled target data from a mixture of positive and negative class with a goal of learning a positive-versus-negative classifier on target. We now introduce the OSLS setting:

**Definition 1** (Open set label shift). *Define $\mathcal{Y}_t = \mathcal{Y}$ and $\mathcal{Y}_s = \mathcal{Y} \backslash \{k + 1\}$. Under OSLS, the label distribution among source classes $\mathcal{Y}_s$ may change but the class conditional $p(x|y)$ for those classes remain invariant between source and target, and the target domain may contain a novel class, i.e.,*

$$p_s(x|y = j) = p_t(x|y = j) \quad \forall j \in \mathcal{Y}_s \qquad and \qquad p_s(y = k + 1) = 0 \,. \tag{1}$$

*Additionally, we have non-zero support for all $k$ (previously-seen) labels in the source distribution, i.e., for all $y \in \mathcal{Y}_s$, $p_s(y) \geqslant c$ for some $c > 0$.*

Note that the label shift and PU learning problems can be obtained as special cases of OSLS. When no novel class is observed in target (i.e., when $p_t(y = k + 1) = 0$), we recover the label shift problem, and when we observe only one class in source (i.e., when $k = 1$), the OSLS problem reduces to PU learning. Under OSLS, our goal naturally breaks down into two tasks: (i) estimate the target label marginal $p_t(y)$ for each class $y \in \mathcal{Y}$; (ii) train a classifier $f : \mathcal{X} \to \Delta^k$ to approximate $p_t(y|x)$.

## 4 Identifiablity of OSLS

We now introduce conditions for OSLS, under which the solution is identifiable. Throughout the section, we will assume access to population distribution for labeled source data and unlabeled target data, i.e., $p_s(x, y)$ and $p_t(x)$ is given. To keep the discussion simple, we assume finite input domain $\mathcal{X}$ which can then be relaxed to continuous inputs. We relegate proofs to App. B.

We first make a connection between target label marginal $p_t(y)$ estimation and learning the target classifier $p_t(y|x)$ showing that recovering $p_t(y)$ is enough to identify $p_t(y|x)$. In population, given access to $p_t(y)$, the class conditional $p_t(x|y = k + 1)$ can be obtained in closed form as $\left(p_t(x) - \sum_{j=1}^k p_t(y = j) p_s(x|y = j)\right) / p_t(y = k + 1)$. We can then apply Bayes rule to obtain $p_t(y|x)$. Henceforth, we will focus our discussion on identifiability of $p_t(y)$ which implies identifiability of $p_t(y|x)$. In following proposition, we present *weak positivity*, a necessary condition for $p_t(y)$ to be identifiable.

**Proposition 1** (Necessary conditions). *Assume $p_t(y) > 0$ for all $y \in \mathcal{Y}_t$. Then $p_t(y)$ is identified only if $p_t(x|y = k + 1)$ and $p_s(x|y)$ for all $y \in \mathcal{Y}_s$ satisfy weak positivity, i.e., there must exists a subdomain $X_{wp} \subset X$ such that:*

   *(i) $p_t(X_{wp}|y = k + 1) = 0$; and*

   *(ii) the matrix $[p_s(x|y)]_{x \in X_{wp}, y \in \mathcal{Y}_s}$ is full column-rank.*

Intuitively, Proposition 1 states that if the target marginal doesn't lie on the vertex of the simplex $\Delta^k$, then their must exist a subdomain $X_{wp}$ where the support of novel class is zero and within $X_{wp}$, $p_t(y)$ for source classes is identifiable. While it may seem that existence of a subdomain $X_{wp}$ is enough, we show that for the OSLS problem, existence doesn't imply uniqueness. In App. B.1, we construct an example, where the weak positivity condition is not sufficient. In that example, we show that there can exist two subdomains $X_{wp}$ and $X'_{wp}$ satisfying weak positivity, both of which lead to separate solutions for $p_t(y)$. Next, we extend weak positivity to two stronger conditions, either of which (alone) implies identifiability.

**Proposition 2** (Sufficient conditions). *The target marginal $p_t(y)$ is identified if for all $y \in \mathcal{Y} \backslash \{k + 1\}$, $p_t(x|y = k + 1)$ and $p_s(x|y)$ satisfy either:*

   *(i) Strong positivity, i.e., there exists $X_{sp} \subset \mathcal{X}$ such that $p_t(X_{sp}|y = k + 1) = 0$ and the matrix $[p_s(x|y)]_{x \in X_{sp}, y \in \mathcal{Y}_s}$ is full-rank and diagonal; or*

*(ii) Separability, i.e., there exists $X_{sep} \subset \mathcal{X}$, such that $p_t(X_{sep}|y = k + 1) = 0$, $p_s(X_{sep}) = 1$, and the matrix $[p_s(x|y)]_{x \in X_{sep}, y \in \mathcal{Y}_s}$ is full column-rank.*

Strong positivity generalizes the irreducibility condition [10], which is sufficient for identifiability under PU learning, to $k$ PU learning problems. Note that while the two conditions in Proposition 2 overlap, they cover independent set of OSLS problems. Informally, strong positivity extends weak positivity by making an additional assumption that the matrix formed by $p(x|y)$ on inputs in $X_{\text{wp}}$ is diagonal and the separability assumption extends the weak positivity condition to the full input domain of source classes instead of just $X_{\text{wp}}$. Both of these conditions identify a support region of $\mathcal{X}$ which purely belongs to source classes where we can either individually estimate the proportion of each source classes (i.e., under strong positivity) or jointly estimate the proportion (i.e., under separability).

To extend our identifiability conditions for continuous distributions, the linear independence conditions on the matrix $[p_s(x|y)]_{x \in X_{\text{sep}}, y \in \mathcal{Y}_s}$ has the undesirable property of being sensitive to changes on sets of measure zero. We may introduce stronger notions of linear independence as in Lemma 1 of Garg et al. [27]. We discuss this in App. B.2.

## 5    Reduction of OSLS to $k$ PU Problems

Under the strong positivity condition, the OSLS problem can be broken down into $k$ PU problems as follows: By treating a given source class $y_j \in \mathcal{Y}_s$ as *positive* and grouping all other classes together as *negative* we observe that the unlabeled target data is then a mixture of data from the positive and negative classes. This yields a PU learning problem and the corresponding mixture proportion is the fraction $p_t(y = j)$ (proportion of class $y_j$) among the target data. By iterating this process for all source classes, we can solve for the entire target label marginal $p_t(y)$. Thus, OSLS reduces to $k$ instances of PU learning problem. Formally, note that $p_t(x)$ can be written as:

$$p_t(x) = p_t(y = j)p_s(x|y = j) + (1 - p_t(y = j)) \left( \sum_{i \in \mathcal{Y} \setminus \{j\}} \frac{p_t(y = i)}{1 - p_t(y = j)} p_s(x|y = i) \right) , \quad (2)$$

individually for all $j \in \mathcal{Y}_s$. By repeating this reduction for all classes, we obtain $k$ separate PU learning problems. Hence, a natural choice is to leverage this structure and solve $k$ PU problems to solve the original OSLS problem. In particular, for each class $j \in \mathcal{Y}_s$, we can first estimate its prevalence $\widehat{p}_t(y = j)$ in the unlabeled target. Then the target marginal for the novel class is given by $\widehat{p}_t(y = k + 1) = 1 - \sum_{i=1}^{k} \widehat{p}_t(y = i)$. Similarly, for classification, we can train $k$ PU learning classifiers $f_i$, where $f_i$ is trained to classify a source class $i$ versus others in target. An example is classified as belonging to the class $y = k + 1$, if it rejected by all classifiers $f_i$ as other in target. We explain this procedure more formally in App. A.1.

This reduction has been mentioned in past work [60, 72]. However, to the best of our knowledge, no previous work has empirically investigated both classification and target label marginal estimation jointly. Sanderson and Scott [60] focuses only on target marginal estimation for tabular datasets and Xu et al. [72] assumes that the target marginal is known and only trains $k$ separate PU classifiers.

In our work, we perform the first large scale experiments to evaluate efficacy of the reduction of the OSLS problem to $k$-PU problems. With plugin state-of-the-art PU learning algorithms, we observe that this naive reduction doesn't scale to datasets with large number of classes because of error accumulation in each of the $k$ MPEs and $k$ one-versus-other PU classifiers. To mitigate the error accumulation problem, we propose the PULSE framework in the next section.

## 6    The PULSE Framework for OSLS

We begin with presenting our framework for OSLS problem under strong positivity condition. First, we explain the structure of OSLS that we leverage in PULSE framework and then elaborate design decisions we make to exploit the identified structure.

**Overview of PULSE framework**    Rather than simply dividing each OSLS instance into $k$ PU problems, we exploit the joint structure of the problem to obtain a *single* PU learning problem. To begin, we note that if only we could apply a *label shift correction* to source, i.e., re-sample source classes according to their relative proportion in the target data, then we could subsequently consider

---

**Algorithm 1** Positive and Unlabeled learning post Label Shift Estimation (PULSE) framework

---

**input** : Labeled source data $\{\mathbf{X}^S, \mathbf{y}^S\}$ and unlabeled target samples $\mathbf{X}^T$.
 1: Randomly split data into training $\{\mathbf{X}_1^S, \mathbf{y}_1^S\}$, $\mathbf{X}_1^T$ and hold out partition $\{\mathbf{X}_2^S, \mathbf{y}_2^S\}$, $\mathbf{X}_2^T$.
 2: Train a source classifier $f_s$ on labeled source data $\{\mathbf{X}_1^S, \mathbf{y}_1^S\}$.
 3: Estimate label shift $\widehat{p}_t'(y = j) = \dfrac{\widehat{p}_t(y = j)}{\sum_{k \in \mathcal{Y}_s} \widehat{p}_t(y = k)}$ using Algorithm 2 and hence importance ratios $\widehat{w}(j)$ among source classes $j \in \mathcal{Y}_s$.
 4: Re-sample training source data according to label distribution $\widehat{p}_t'$ to get $\{\widetilde{\mathbf{X}}_1^S, \widetilde{\mathbf{y}}_1^S\}$ and $\{\widetilde{\mathbf{X}}_2^S, \widetilde{\mathbf{y}}_2^S\}$.
 5: Using Algorithm 3, train a discriminator $f_d$ and estimate novel class fraction $\widehat{p}_t(y = k + 1)$.
 6: Assign $[f_t(x)]_j = (f_d(x)) \dfrac{\widehat{w}(j) \cdot [f_s(x)]_j}{\sum_{k \in \mathcal{Y}_s} \widehat{w}(k) \cdot [f_s(x)]_k}$ for all $j \in \mathcal{Y}_s$ and $[f_t(x)]_{k+1} = 1 - f_d(x)$. And for all $j \in \mathcal{Y}_s$, assign $\widehat{p}_t(y = j) = (1 - \widehat{p}_t(y = k + 1)) \cdot \widehat{p}_t'(y = j)$.
**output** : Target marginal estimate $\widehat{p}_t \in \Delta^k$ and target classifier $f_t(\cdot) \in \Delta^k$.

---

the unlabeled target data as a mixture of (i) the (reweighted) source distribution; and (ii) the novel class distribution (i.e., $p_t(x|y = k + 1)$). Formally, we have

$$p_t(x) = \sum_{j \in \mathcal{Y}_t} p_t(y = j) p_t(x|y = j) = \sum_{j \in \mathcal{Y}_s} \frac{p_t(y = j)}{p_s(y = j)} p_s(x, y = j) + p_t(x|y = k + 1) p_t(y = k + 1)$$

$$= (1 - p_t(y = k + 1)) p_s'(x) + p_t(y = k + 1) p_t(x|y = k + 1), \qquad (3)$$

where $p_s'(x)$ is the label-shift-corrected source distribution, i.e., $p_s'(x) = \sum_{j \in \mathcal{Y}_s} w(j) p_s(x, y = j)$, where $w(j) = \left( p_t(y = j) / \sum_k p_t(y = k) \right) / p_s(y = j)$ for all $j \in \mathcal{Y}_s$. Intuitively, $p_t'(j) = p_t(y = j) / \sum_k p_t(y = k)$ is re-normalized label distribution in target among source classes and $w(j)$'s are the importance weights. Hence, after applying a label shift correction to the source distribution $p_s'(x)$, we have reduced the OSLS problem to a *single* PU learning problem, where $p_s'(x)$ plays the part of the positive distribution and $p_t(x|y = k + 1)$ acts as negative distribution with mixture coefficients $1 - p_t(y = k + 1)$ and $p_t(y = k + 1)$ respectively. We now discuss our methods (i) to estimate the importance ratios $w(y)$; and (ii) to tackle the PU learning instance obtained from OSLS.

**Label shift correction: Target marginal estimation among source classes**   While traditional methods for estimating label shift breakdown in high dimensional settings [78], recent methods exploit black-box classifiers to avoid the curse of dimensionality [45, 4, 1]. However, these recent techniques require overlapping label distributions, and a direct application would require demarcation of samples from $p_s'(x)$ sub-population in target, creating a cyclic dependency. Instead, to estimate the relative proportion of previously seen classes in target, we leverage the $k$ PU reduction described in Sec. 5 with two crucial distinctions. First, we normalize the obtained estimates of fraction previously seen classes to obtain the relative proportions in $p_s'(y)$. In particular, we do not leverage the estimates of previously seen class proportions in target to directly estimate the proportion of novel class which avoids issues due to error accumulation. Second, we exploit a $k$-way source classifier $f_s$ trained on labeled source data instead of training $k$ one-versus-other PU classifiers. We tailor the recently proposed Best Bin Estimation (BBE) technique from Garg et al. [29]. We describe the modified BBE procedure in App. C (Algorithm 2). After estimating the relative fraction of source classes in target (i.e., $\widehat{p}_t'(j) = \widehat{p}_t(y = j) / \sum_{k \in \mathcal{Y}_s} \widehat{p}_t(y = k)$ for all $j \in \mathcal{Y}_s$), we re-sample the source data according to $\widehat{p}_t'(y)$ to mimic samples from distribution $p_s'(x)$.

**PU Learning: Separating the novel class from previously seen classes**   After obtaining a PU learning problem instance, we resort to PU learning techniques to (i) estimate the fraction of novel class $p_t(y = k+1)$; and (ii) learn a binary classifier $f_d(x)$ to discriminate between label shift corrected source $p_s'(x)$ and novel class $p_t(x|y = k + 1)$. With traditional methods for PU learning involving domain discrimination, over-parameterized models can memorize the positive instances in unlabeled, assigning them confidently to the negative class, which can severely hurt generalization on PN data [38, 29]. Rather, we employ Conditional Value Ignoring Risk (CVIR) loss proposed in Garg et al. [29] which was shown to outperform alternative approaches. First, we estimate the proportion of novel class $\widehat{p}_t(y = k + 1)$ with BBE. Next, given an estimate $\widehat{p}_t(y = k + 1)$, CVIR objective discards the highest loss $(1 - \widehat{p}_t(y = k + 1))$ fraction of examples on each training epoch, removing the incentive to overfit to the examples from $p_s'(x)$. Consequently, we employ the iterative procedure that alternates

between estimating the prevalence of novel class $\widehat{p}_t(y = k + 1)$ (with BBE) and minimizing the CVIR loss with estimated fraction of novel class. We detail this procedure in App. C (Algorithm 3).

**Combining PU learning and label shift correction**   Finally, to obtain a $(k+1)$-way classifier $f_t(x)$ on target we combine discriminator $f_d$ and source classifier $f_s$ with importance-reweighted label shift correction. In particular, for all $j \in \mathcal{Y}_s$, $[f_t(x)]_j = (f_d(x))\frac{w(j) \cdot [f_s(x)]_j}{\sum_{k \in \mathcal{Y}_s} w(k) \cdot [f_s(x)]_k}$ and $[f_t(x)]_{k+1} = 1 - f_d(x)$. Overall, our approach outlined in Algorithm 1 proceeds as follows: First, we estimate the label shift among previously seen classes. Then we employ importance re-weighting of source data to formulate a single PU learning problem to estimate the fraction of novel class $\widehat{p}_t(y = k + 1)$ and to learn a discriminator $f_d$ for the novel class. Combining discriminator and label shift corrected source classifier we get $(k + 1)$-way target classifier. We analyse crucial steps in PULSE in Sec. 8.

Our ideas for PULSE framework can be extended to separability condition since (3) continues to hold. However, in our initial experiments, we observe that techniques proposed under strong positivity were empirically stable and outperform methods developed under separability. This is intuitive for many benchmark datasets where it is natural to assume that for each class there exists a subdomain that only belongs to that class. We describe this in more detail in App. C.1.

## 7    Experiments

**Baselines**   We compare PULSE with several popular methods from OSDA literature. While these methods are not specifically proposed for OSLS, they are introduced for the more general OSDA problem. In particular, we make comparions with DANCE [59], UAN [73], CMU [25], STA [46], Backprop-ODA (or BODA) [58]. We use the open source implementation available at `https://github.com/thuml`. For alternative baselines, we experiment with source classifier directly deployed on the target data which may contain novel class and label shift among source classes (referred to as *source-only*). We also train a domain discriminator classifier for source versus target (referred to as *domain disc.*). This is adaptation of PU learning baseline[24] which assumes no label shift among source classes. Finally, per the reduction presented in Sec. 5, we train $k$ PU classifiers (referred to as *k-PU*). We include detailed description of each method in App. F.1.

**Datasets**   We conduct experiments with seven benchmark classification datasets across vision, natural language, biology and medicine. For each dataset, we simulate an OSLS problem as described in next paragraph. For vision, we use CIFAR10, CIFAR100  [40] and Entity30 [61]. For language, we experiment with Newsgroups-20 (`http://qwone.com/~jason/20Newsgroups/`) dataset. Additionally, inspired by applications of OSLS in biology and medicine, we experiment with Tabula Muris [17] (Gene Ontology prediction), Dermnet (skin disease prediction `https://dermnetnz.org/`), and BreakHis [66] (tumor cell classification). These datasets span language, image and table modalities. We provide interpretation of OSLS problem for each dataset along with other details in App. F.2.

**OSLS Setup**   To simulate an OSLS problem, we experiment with different fraction of novel class prevalence, source label distribution, and target label distribution. We randomly choose classes that constitute the novel target class. After randomly choosing source and novel classes, we first split the training data from each source class randomly into two partitions. This creates a random label distribution for shared classes among source and target. We then club novel classes to assign them a new class (i.e. $k + 1$). Finally, we throw away labels for the target data to obtain an unsupervised DA problem. We repeat the same process on iid hold out data to obtain validation data with no target labels.

**Training and Evaluation**   We use Resnet18 [33] for CIFAR10, CIFAR100, and Entity30. For newsgroups, we use a convolutional architecture. For Tabular Muris and MNIST, we use a fully connected MLP. For Dermnet and BreakHis, we use Resnet-50. For all methods, we use the same backbone for discriminator and source classifier. For kPU, we use a separate final layer for each class with the same backbone. We use default hyperparameters for all methods. For OSDA methods, we use default method specific hyperparameters introduced in their works. Since OSDA methods do not estimate the prevalence of novel class explicitly, we use the fraction of examples predicted in class $k + 1$ as a surrogate. We train models till the performance on validation source data (labeled) ceases to increase. Unlike OSDA methods, note that we do not use early stopping based on performance on held-out labeled target data. To evaluate classification performance, we report target accuracy on all classes, seen classes and the novel class. For novel class prevalence estimation, we report absolute difference between the true and estimated marginal. We open-source our code and by simply

Table 1: *Comparison of PULSE with other methods.* Across all datasets, PULSE outperforms alternatives for both target classification and novel class prevalence estimation. Acc (All) is target accuracy, Acc (Seen) is target accuracy on examples from previously seen classes, and Acc (Novel) is recall for novel examples. MPE (Novel) is absolute error for novel prevalence estimation. Results reported by averaging across 3 seeds. Detailed results for each dataset with all methods in App. F.4.

| Method | CIFAR-10 | | | | CIFAR-100 | | | |
|---|---|---|---|---|---|---|---|---|
| | Acc (All) | Acc (Seen) | Acc (Novel) | MPE (Novel) | Acc (All) | Acc (Seen) | Acc (Novel) | MPE (Novel) |
| Source-Only | 67.1 | 87.0 | - | - | 46.6 | 66.4 | - | - |
| UAN [73] | 15.4 | 19.7 | 25.2 | 0.214 | 18.1 | 40.6 | 14.8 | 0.133 |
| BODA [58] | 63.1 | 66.2 | 42.0 | 0.162 | 36.1 | 17.7 | 81.6 | 0.41 |
| DANCE [59] | 70.4 | 85.5 | 14.5 | 0.174 | 47.3 | 66.4 | 1.2 | 0.28 |
| STA [46] | 57.9 | 69.6 | 14.9 | 0.124 | 42.6 | 48.5 | 34.8 | 0.14 |
| CMU [25] | 62.1 | 77.9 | 41.2 | 0.183 | 35.4 | 46.0 | 15.5 | 0.161 |
| Domain Disc. [24] | 47.4 | 87.0 | 30.6 | 0.331 | 45.8 | 66.5 | 39.1 | **0.046** |
| $k$-PU | 83.6 | 79.4 | **98.9** | 0.036 | 36.3 | 22.6 | **99.1** | 0.298 |
| PULSE (Ours) | **86.1** | **91.8** | 88.4 | **0.008** | **63.4** | **67.2** | 63.5 | 0.078 |

| Method | Entity30 | | Newsgroups20 | | Tabula Muris | | BreakHis | | DermNet | |
|---|---|---|---|---|---|---|---|---|---|---|
| | Acc (All) | MPE (Novel) | Acc (All) | MPE (Novel) | Acc (All) | MPE (Novel) | Acc (All) | MPE (Novel) | Acc (All) | MPE (Novel) |
| Source-Only | 32.0 | - | 39.3 | - | 33.8 | - | 70.0 | - | 41.4 | - |
| BODA [58] | 42.2 | 0.189 | 43.4 | 0.16 | 76.5 | 0.079 | 71.5 | 0.077 | 43.8 | 0.207 |
| Domain Disc. | 43.2 | 0.135 | 50.9 | 0.176 | 73.0 | 0.071 | 56.5 | 0.091 | 40.6 | 0.083 |
| $k$-PU | 50.7 | 0.394 | 52.1 | 0.373 | 85.9 | 0.307 | 75.6 | **0.059** | 46.0 | 0.313 |
| PULSE (Ours) | **58.0** | **0.054** | **62.2** | **0.061** | **87.8** | **0.058** | **79.1** | **0.054** | **48.9** | **0.043** |

changing a single config file, new OSLS setups can be generated and experimented with. We provide precise details about hyperparameters, OSLS setup for each dataset and code in App. F.3.

**Results** Across different datasets, we observe that PULSE consistently outperforms other methods for the target classification and novel prevalence estimation (Table 1). For detection of novel classes (Acc (Novel) column), kPU achieves superior performance as compared to alternative approaches because of its bias to default to $(k+1)^{\text{th}}$ class. This is evident by the sharp decrease in performance on previously seen classes. For each dataset, we plot evolution of performance with training in App. F.4. We observe more stability in performance of PULSE as compared to other methods.

We observe that with default hyperparameters, popular OSDA methods significantly under perform as compared to PULSE. We hypothesize that the primary reasons underlying the poor performance of OSDA methods are (i) the heuristics employed to detect novel classes; and (ii) loss functions incorporated to improve alignment between examples from common classes in source and target. To detect novel classes, a standard heuristic employed in popular OSDA methods involves thresholding uncertainty estimates (e.g., prediction entropy, softmax confidence [73, 25, 59]) at a predefined threshold $\kappa$. However, a fixed $\kappa$, may not for different datasets and different fractions of the novel class. In App. F.5, we ablate by (i) removing loss function terms incorporated with an aim to improve source target alignment; and (ii) vary threshold $\kappa$ and show improvements in performance of these methods. In contrast, our two-stage method PULSE, first estimates the fraction of novel class which then guides the classification of novel class versus previously seen classes avoiding the need to guess $\kappa$.

**Ablations** Different datasets, in our setup span different fraction of novel class prevalence ranging from 0.22 (in CIFAR10) to 0.64 (in Tabula Muris). For each dataset, we perform more ablations on the novel class proportion in App. F.6. For kPU and PULSE, in the main paper, we include results with BBE and CVIR [29]. In App. F.8, we perform experiments with alternative PU learning approaches

and highlight the superiority of BBE and CVIR over other methods. Moreover, since we have access to unlabeled target data, we experiment with SimCLR [15] pre-training on the mixture of unlabeled source and target dataset. We include setup details and results in App. F.7. While pre-trained backbone architecture improves performance for all methods, PULSE continues to dominate other methods.

## 8 Analysis of PULSE Framework

In this section, we analyse key steps of our PULSE procedure for target label marginal estimation (Step 3, 5 Algorithm 1) and learning the domain discriminator classifier (Step 5, Algorithm 1). Due to space constraints, we present informal results here and relegate formal statements and proofs to App. D.

**Theoretical analysis for target marginal estimation**     Building on BBE results from Garg et al. [29], we present finite sample results for target label marginal estimation. When the data satisfies strong positivity, we observe that source classifiers often exhibit a threshold $c_y$ on softmax output of each class $y \in \mathcal{Y}_s$ above which the *top bin* (i.e., $[c_y, 1]$) contains mostly examples from that class $y$. We give empirical evidence to this claim in App. D.1. Then, we show that the existence of (nearly) pure top bin for each class in $f_s$ is sufficient for Step 3 in Algorithm 1 to produce (nearly) consistent estimates:

**Theorem 1** (Informal). *Assume that for each class $y \in \mathcal{Y}_s$, there exists a threshold $c_y$ such that for the classifier $f_s$, if $[f_s(x)]_y > c_y$ for any $x$ then the true label for that sample $x$ is $y$. Then, we have*
$$\|\widehat{p}_t - p_t\|_1 \leqslant \mathcal{O}\left(\sqrt{k^3 \log(4k/\delta)/n} + \sqrt{k^2 \log(4k/\delta)/m}\right).$$

The proof technique simply builds on the proof of Theorem 1 in Garg et al. [29]. By assuming that we recover close to ground truth label marginal for source classes, we can also extend the above analysis to Step 5 of Algorithm 1 to show convergence of estimate $\widehat{p}_t(y = k+1)$ to true prevalence $p_t(y = k+1)$. We discuss this further in App. D.3.

**Theoretical analysis of CVIR in population**     While the CVIR loss was proposed in Garg et al. [29], no analysis was provided for convergence of the iterative gradient descent procedure. In our work, we show that in population on a separable Gaussian dataset, CVIR will recover the optimal classifier.

We consider a binary classification problem where we have access to positive distribution (i.e., $p_p$), unlabeled distribution (i.e., $p_u := \alpha p_p + (1 - \alpha)p_n$), and mixture coefficient $\alpha$. Making a parallel connection to Step 5 of PULSE, positive distribution $p_p$ here refers to the label shift corrected source distribution $p'_s$ and $p_u$ refers to $p_t = p_t(y = k+1)p_t(x|y = k+1) + (1 - p_t(y = k+1))p'_s(x)$. Our goal is to recover the classifier that discriminates $p_p$ versus $p_n$ (parallel $p'_s$ versus $p_t(\cdot|y = k+1)$).

First we introduce some notation. For a classifier $f$ and loss function $\ell$ (i.e., logistic loss), define $\text{VIR}_\alpha(f) = \inf\{\tau \in \mathbb{R} : \text{P}_{x \sim p_u}(\ell(x, -1; f) \leqslant \tau) \geqslant 1 - \alpha\}$. Intuitively, $\text{VIR}_\alpha(f)$ identifies a threshold $\tau$ to capture bottom $1 - \alpha$ fraction of the loss $\ell(x, -1)$ for points $x$ sampled from $p_u$. Additionally, define CVIR loss as $\mathcal{L}(f, w) = \alpha\mathbb{E}_{p_p}\left[\ell(x, 1; f)\right] + \mathbb{E}_{p_u}\left[w(x)\ell(x, -1; f)\right]$ for classifier $f$ and some weights $w(x) \in \{0, 1\}$. Formally, given a classifier $f_t$ at an iterate $t$, CVIR procedure proceeds as follows:

$$w_t(x) = \mathbb{I}\left[\ell(x, -1; f_t) \leqslant \text{VIR}_\alpha(f_t)\right], \tag{4}$$
$$f_{t+1} = f_t - \eta\nabla\mathcal{L}_f(f_t, w_t). \tag{5}$$

We assume that $x$ are drawn from two half multivariate Gaussian with mean zero and identity covariance, i.e., $x \sim p_p \Leftrightarrow x = \gamma_0\theta_{\text{opt}} + z | \theta_{\text{opt}}^T z \geqslant 0$, and $x \sim p_n \Leftrightarrow x = -\gamma_0\theta_{\text{opt}} + z | \theta_{\text{opt}}^T z < 0$, where $z \sim \mathcal{N}(0, I_d)$. Here $\gamma_0$ is the margin and $\theta_{\text{opt}} \in \mathbb{R}^d$ is the true separator. Here, we have access to distribution $p_p, p_u = \alpha p_p + (1 - \alpha)p_n$, and the true proportion $\alpha$.

**Theorem 2** (Informal). *In the data setup detailed above, a linear classifier $f(x; \theta) = \sigma\left(\theta^T x\right)$ trained with CVIR procedure as in* (4)-(5) *will converge to an optimal positive versus negative classifier.*

The proof uses a key idea that for any classifier $\theta$ not separating positive and negative data perfectly, the gradient in (5) is non-zero. Hence, convergence of the CVIR procedure (implied by smoothness of CVIR loss) implies converge to an optimal classifier. For separable datasets in general, we can extend the above analysis with some modifications to the CVIR procedure. We discuss this in App. D.4.

**Empirical investigation in overparameterized models** As noted in our ablation experiments and in Garg et al. [29], domain discriminator trained with CVIR outperforms classifiers trained with other consistent objectives (nnPU [38] and uPU [21]). While the above analysis highlights consistency of CVIR procedure in population, it doesn't capture the observed empirical efficacy of CVIR over alternative methods in overparameterized models. In the Gaussian setup described above, we train overparameterized linear models to compare CVIR with other methods. We discuss precise experiments and results in App. E, but highlight the key takeaway here. First, we observe that when a classifier is trained to distinguish positive and unlabeled data, *early learning* happens [47, 3, 28], i.e., during the initial phase of learning classifier learns to classify positives in unlabeled correctly as positives. Next, we show that post early learning rejection of large fraction of positives from unlabeled training in equation (4) crucially helps CVIR.

## 9 Conclusion

In this work, we introduce OSLS a well-posed instantiation of OSDA that subsumes label shift and PU learning into a framework for learning adaptive classifiers. We presented identifiability conditions for OSLS and proposed PULSE, a simple and effective approach to tackle the OSLS problem. Moreover, our extensive experiments demonstrate efficacy of PULSE over popular OSDA alternatives when the OSLS assumptions are met. We would like to highlight the brittle nature of benchmark driven progress in OSDA and hope that our work can help to stimulate more solid foundations and enable systematic progress in this area. Finally, we hope that our open source code and benchmarks will foster further progress on OSLS.

### 9.1 Limitations and Future Work

Here, we discuss limitations of the PULSE framework. First, to estimate the relative label shift among source classes in target, we leverage k-PU reductions with several modifications. While we reduce the issues due to overestimation bias by re-normalizing the label marginal among source classes in target, in future, we may hope to replace this heuristic step to directly estimate the joint target marginal.

Second, since our methods use CVIR and BBE sub-routines, failure of these methods can lead to failure of PULSE. For example, efficacy of BBE relies on the existence of an almost pure top bin in the classifier output space. While this property seems to be satisfied across different datasets spanning different modalities and applications, failure to identify an almost pure top bin can degrade the performance of BBE and hence, our PULSE framework.

In future work, we also hope to bridge the gap between the necessary and sufficient identifiability conditions. While we empirically investigate reasons for CVIR's efficacy in overparameterized models, we aim to extend our theory to overparameterized settings in future. In our work, we strictly operate under the OSLS settings, where we performed semi-synthetic experiments on vision, language and tabular datasets. In future, it will be interesting to experiment with our PULSE procedure in relaxed settings where $p(x|y)$ may shift in some natural-seeming ways from source to target.

## Acknowledgments and Disclosure of Funding

We thank Jennifer Hsia for initial discussion on the OSLS problem. We also thank Euxhen Hasanaj for suggesting Biology datasets. SG acknowledges Amazon Graduate Fellowship for their support. SB acknowledges funding from the NSF grants DMS-1713003, DMS-2113684 and CIF-1763734, as well as Amazon AI and a Google Research Scholar Award. ZL acknowledges Amazon AI, Salesforce Research, Facebook, UPMC, Abridge, the PwC Center, the Block Center, the Center for Machine Learning and Health, and the CMU Software Engineering Institute (SEI) via Department of Defense contract FA8702-15-D-0002, for their generous support of ACMI Lab's research on machine learning under distribution shift.

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
