# Supplementary Materials for Domain Adaptation under Open Set Label Shift

## A  Preliminaries

**Domain adaptation under label shift**  Under label shift, we observe data from $k$ classes in both source and target where the conditional distribution remain invariant (i.e., $p_s(x|y) = p_t(x|y)$ for all classes $y \in [1, k]$) but the target label marginal may change (i.e., $p_t(y) \neq p_s(y)$). Additionally, for all classes in source have a non-zero support , i.e., for all $y \in [1, k]$, $p_s(y) \geqslant c$, where $c > 0$. Here, given labeled source data and unlabeled target data our tasks are: (i) estimate the shift in label distribution, i.e., $p_t(y)$ for all $y \in [1, k]$; (ii) train a classifier for the target domain $f_t$ to approximate $p_t(y|x)$.

One common approach to label shift involves estimating the importance ratios $p_t(y)/p_s(y)$ by leveraging a blackbox classifier and then employing re-sampling of source data or importance re-weighted ERM on source to obtain a classifier for the target domain [45, 4, 1].

**PU learning**  Under PU learning, we possess labeled source data from a positive class ($p_p$) and unlabeled target data from $p_u = \alpha p_p + (1 - \alpha)p_n$ a mixture of positive and negative class ($p_n$). Our goals naturally break down in to two tasks: (i) MPE, determining the fraction of positives $p_p$ in $p_u$ and (ii) PU classification, learning a positive-versus-negative classifier on target.

Note that given access to population of positives and unlabeled, $\alpha$ can be estimated as $\min_x p_u(x)/p_p(x)$. Next, we briefly discuss recent methods for MPE that operate in the classifier output space to avoid curse of dimensionality:

(i) **EN:** Given a domain discriminator classifier $f_d$ trained to discriminate between positive and unlabeled, Elkan and Noto [24] proposed the following estimator: $\sum_{x_i \in X_p} f_d(x_i)/\sum_{x_i \in X_u} f_d(x_i)$ where $X_p$ is the set of positive examples and $X_u$ is the set of unlabeled examples.

(ii) **DEDPUL:** Given a domain discriminator classifier $f_d$, Ivanov [35] proposed an estimator that leverages density of the data in the output space of the classifier $f_d$ to directly estimate $\min p_u(f(x))/p_p(f(x))$.

(iii) **BBE:** BBE [29] identifies a threshold on probability scores assigned by the classifier $f_d$ such that by estimating the ratio between the fractions of positive and unlabeled points receiving scores above the threshold, we obtain proportion of positives in unlabeled.

After obtaining an estimate for mixture proportion $\alpha$, following methods can be employed for PU classification:

(i) **Domain Discriminator:** Given positive and unlabeled data, Elkan and Noto [24] trained a classifier $f_d$ to discriminator between them. To make a prediction on test point from unlabeled data, we can then use Bayes rule to obtain the following transformation on probabilistic output of the domain discriminator: $f = \alpha \left( \frac{m}{n} \right) \left( \frac{f_d(x)}{1 - f_d(x)} \right)$, where $n$ and $m$ are the number of positives and unlabeled examples used to train $f_d$ [24].

(ii) **uPU:** Du Plessis et al. [21] proposed an unbiased loss estimator for positive versus negative training. In particular, since $p_u = \alpha p_p + (1 - \alpha)p_n$, the loss on negative examples $\mathbb{E}_{p_n} [\ell(f(x); -1)]$ can be estimated as:

$$\mathbb{E}_{p_n} [\ell(f(x); -1)] = \frac{1}{1 - \alpha} \left[ \mathbb{E}_{p_u} [\ell(f(x); -1)] - \alpha \mathbb{E}_{p_p} [\ell(f(x); -1)] \right] . \quad (6)$$

Thus, a classifier can be trained with the following uPU loss:

$$\mathcal{L}_{\text{uPU}}(f) = \alpha \mathbb{E}_{p_p} [\ell(f(x); +1)] + \mathbb{E}_{p_u} [\ell(f(x); -1)] - \alpha \mathbb{E}_{p_p} [\ell(f(x); -1)] . \quad (7)$$

(iii) **nnPU:** While unbiased losses exist that estimate the PvN loss given PU data and the mixture proportion $\alpha$, this unbiasedness only holds before the loss is optimized, and becomes

ineffective with powerful deep learning models capable of memorization. Kiryo et al. [38] proposed the following non-negative regularization for unbiased PU learning:

$$\mathcal{L}_{\text{nnPU}}(f) = \alpha \mathbb{E}_{p_p}\left[\ell(f(x); +1)\right] + \max\left\{\mathbb{E}_{p_u}\left[\ell(f(x); -1)\right] - \alpha \mathbb{E}_{p_p}\left[\ell(f(x); -1)\right], 0\right\}. \tag{8}$$

(iv) **CVIR:** Garg et al. [29] proposed CVIR objective, which discards the highest loss $\alpha$ fraction of unlabeled examples on each training epoch, removing the incentive to overfit to the unlabeled positive examples. CVIR loss is defined as

$$\mathcal{L}_{\text{CVIR}}(f) = \alpha \mathbb{E}_{p_p}\left[\ell(x, 1; f)\right] + \mathbb{E}_{p_u}\left[w(x)\ell(x, -1; f)\right], \tag{9}$$

where weights $w(x) = \mathbb{I}\left[\ell(x, -1; f) \leqslant \text{VIR}_\alpha(f)\right]$ for $\text{VIR}_\alpha(f)$ defined as $\text{VIR}_\alpha(f) = \inf\{\tau \in \mathbb{R} : \text{P}_{x \sim p_u}(\ell(x, -1; f) \leqslant \tau) \geqslant 1 - \alpha\}$. Intuitively, $\text{VIR}_\alpha(f)$ identifies a threshold $\tau$ to capture bottom $1 - \alpha$ fraction of the loss $\ell(x, -1)$ for points $x$ sampled from $p_u$.

### A.1 Reduction of OSLS into $k$ PU problems

Under the strong positivity condition, the OSLS problem can be broken down into $k$ PU problems as follows: By treating a given source class $y_j \in \mathcal{Y}_s$ as *positive* and grouping all other classes together as *negative* we observe that the unlabeled target data is then a mixture of data from the positive and negative classes. This yields a PU learning problem and the corresponding mixture proportion gives the fraction $\alpha_j$ of class $y_j$ among the target data. By iterating this process for all source classes, we can solve for the entire target label marginal $p_t(y)$. Thus, OSLS reduces to $k$ instances of PU learning problem. Formally, note that $p_t(x)$ can be written as:

$$p_t(x) = \underbrace{p_t(y = j)}_{\alpha_j} \underbrace{p_s(x|y = j)}_{p_p} + (1 - p_t(y = j)) \underbrace{\left(\sum_{i \in \mathcal{Y} \setminus \{j\}} \frac{p_t(y = i)}{1 - p_t(y = j)} p_s(x|y = i)\right)}_{p_n}, \tag{10}$$

individually for all $j \in \mathcal{Y}_s$. By repeating this reduction for all classes, we obtain $k$ separate PU learning problems. Hence, a natural choice is to leverage this structure and solve $k$ PU problems to solve the original OSLS problem.

In particular, for each class $j \in \mathcal{Y}_s$, we can first estimate its prevalence $\widehat{\alpha}_j$ in the unlabeled target. Then the target marginal for the novel class is given by $\widehat{\alpha}_{k+1} = 1 - \sum_{i=1}^k \widehat{\alpha}_i$. For classification, we can train $k$ PU learning classifiers $f_i$, where $f_i$ is trained to classify a source class $i$ versus others in target. Assuming that each $f_j$ returns a score between $[0, 1]$, during test time, an example $x$ is classified as $f(x)$ given by

$$f(x) = \begin{cases} \arg\max_{j \in \mathcal{Y}_s} f_j(x) & \text{if } \max_{j \in \mathcal{Y}_s} f_j(x) \geqslant 0.5 \\ k + 1 & \text{o.w.} \end{cases} \tag{11}$$

That is, if each classifier classifies the example as belonging to other in unlabeled, then we classify the example as belonging to the class $k + 1$. In our main experiments, to estimate $\alpha_j$ and to train $f_j$ classifiers for all $j \in \mathcal{Y}_s$, we use BBE and CVIR as described before which was shown to outperform alternative approaches in Garg et al. [29]. We ablate with other methods in App. F.8.

Note that mathematically any OSLS problems can be thought of as $k$-PU problems as per (10). However, for identifiablity of each of these PU problems, we need the irreduciblity assumption [7]. Put simply, for individual PU problems defined for source classes $j \in \mathcal{Y}_s$, we need existence of a sub-domain $X_j$ such that we only observe example for that class j in $X_j$. Collectively $X_j$ gives us the $X_{\text{sp}}$ defined in the strong positivity condition.

**Failure due to error-accumulation** While trading off bias with variance, PU learning algorithms tend to over-estimate the mixture proportion [29, 7]. This error incurred due to bias can be mild for a single mixture proportion estimation task but accumulates with increasing number of classes (i.e., $k$). This error accumulation can significantly under-estimate the proportion of novel class when estimated by subtracting the sum of prevalence of source classes in target from 1.

# B  Proofs for identifiability of OSLS

For ease, we re-state Proposition 1 and Proposition 2.

**Proposition 1** (Necessary conditions). *Assume $p_t(y) > 0$ for all $y \in \mathcal{Y}_t$. Then $p_t(y)$ is identified only if $p_t(x|y = k+1)$ and $p_s(x|y)$ for all $y \in \mathcal{Y}_s$ satisfy weak positivity, i.e., there must exists a subdomain $X_{wp} \subset X$ such that:*

*(i) $p_t(X_{wp}|y = k+1) = 0$; and*

*(ii) the matrix $[p_s(x|y)]_{x \in X_{wp}, y \in \mathcal{Y}_s}$ is full column-rank.*

*Proof.* We prove this by contradiction. Assume that there exists a unique solution $p_t(y)$. We will obtain contradiction when both (i) and (ii) don't hold.

First, assume for no subset $X_{wp} \subseteq \mathcal{X}$, we have $[p_s(x|y)]_{x \in X_{wp}, y \in \mathcal{Y}_s}$ as full-rank. Then in that case, we have vectors $[p_s(x|y = j)]_{x \in \mathcal{X}}$ as linearly dependent for $j \in \mathcal{Y}_s$, i.e., there exists $[\alpha_j]_{j \in \mathcal{Y}_s} \in \mathbb{R}^k$ such that $\sum_j \alpha_j p_s(x|y = j) = 0$ for all $x \in \mathcal{X}$. Thus for small enough $\epsilon > 0$, we have infinite solutions of the form $[p_t(y = j) - \epsilon \cdot a_j]_{j \in \mathcal{Y}_s}$.

Hence, there exists $X_{wp} \subseteq \mathcal{X}$ for which we have $[p_s(x|y)]_{x \in X_{wp}, y \in \mathcal{Y}_s}$ as full-rank. Without loss of generality, we assume that $|X_{wp}| = k$. Assume that $p_t(X_{wp}|y = k+1) > 0$, i.e., $[p_t(x|y = k+1)]_{x \in X_{wp}}$ has $l < k$ zero entries. We will now construct another solution for the label marginal $p_t$. For simplicity we denote $A = [p_s(x|y)]_{x \in X_{wp}, y \in \mathcal{Y}_s}$. Consider the vector $v(\gamma) = [p_t(x) - (p_t(y = k+1) - \gamma)p_t(x|y = k+1)]_{x \in X_{wp}}$ for some $\gamma > 0$. Intuitively, when $\gamma = 0$, we have $u = A^{-1}v(0)$ where $u = [p_t(y)]_{y \in \mathcal{Y}_s}$, i.e., we recover the true label marginal corresponding to source classes.

However, since the solution is not at vertex, there exists a small enough $\gamma > 0$ such that $u' = A^{-1}v(\gamma)$ with $\sum_j u'_j \leqslant 1$ and $u'_j \geqslant 0$. Since A is full-rank and $v(\gamma) \neq v(0)$, we have $u' \neq u$. Thus we construct a separate solution with $u'$ as $[p_t(y)]_{y \in \mathcal{Y}_s}$ and $p_t(x) - \sum_{j \in \mathcal{Y}_s} u'_j p_s(x|y = j)$ as $p_t(x|y = k+1)$. Hence, when there exists $X_{wp} \subseteq \mathcal{X}$ for which we have $[p_s(x|y)]_{x \in X_{wp}, y \in \mathcal{Y}_s}$ as full-rank, for uniqueness we obtain a contradiction on the assumption $p_t(X_{wp}|y = k+1) > 0$. $\square$

We now make some comments on the assumption $p_t(y) > 0$ for all $y \in \mathcal{Y}_t$ in Proposition 1. Since, $p_t(y)$ needs to satisfy simplex constraints, if the solution is at a vertex of simplex, then OSLS problem may not require weak positivity. For example, there exists contrived scenarios where $p_s(x|y = j) = p_s(x|y = k)$ for all $j, k \in \mathcal{Y}_s$ and $p_t(x|y = k+1) \neq p_s(x|y = j)$ for all $j \in \mathcal{Y}_s$. Then when $p_t(x) = p_t(x|y = k+1)$, we can uniquely identify the OSLS solution even when weak positivity assumption is not satisfied.

**Proposition 2** (Sufficient conditions). *The target marginal $p_t(y)$ is identified if for all $y \in \mathcal{Y} \backslash \{k+1\}$, $p_t(x|y = k+1)$ and $p_s(x|y)$ satisfy either:*

*(i) Strong positivity, i.e., there exists $X_{sp} \subset \mathcal{X}$ such that $p_t(X_{sp}|y = k+1) = 0$ and the matrix $[p_s(x|y)]_{x \in X_{sp}, y \in \mathcal{Y}_s}$ is full-rank and diagonal; or*

*(ii) Separability, i.e., there exists $X_{sep} \subset \mathcal{X}$, such that $p_t(X_{sep}|y = k+1) = 0$, $p_s(X_{sep}) = 1$, and the matrix $[p_s(x|y)]_{x \in X_{sep}, y \in \mathcal{Y}_s}$ is full column-rank.*

*Proof.* For each condition, we will prove identifiability by constructing the unique solution.

Under strong positivity, for all $j \in \mathcal{Y}_s$ there exists $x \in X_{sp}$ such that $p_t(x|y = k) = 0$ for all $k \in \mathcal{Y}_t \backslash \{j\}$. Set $\alpha_j = \min_{x \in \mathcal{X}, p_s(x|y=j)>0} \frac{p_t(x)}{p_s(x|y=j)}$, for all $j \in \mathcal{Y}_s$. For $x \in X_{sp}$ such that $p_t(x|y = k) = 0$ for all $k \in \mathcal{Y}_t \backslash \{j\}$, we get $\frac{p_t(x)}{p_s(x|y=j)} = p_t(y = j)$ and for all $x' \neq x$, we have $\frac{p_t(x)}{p_s(x|y=j)} \geqslant p_t(y = j)$. Thus, we get $\alpha_j = p_t(y = j)$. Finally, we get $\alpha_{k+1} = 1 - \sum_{j \in \mathcal{Y}_s} \alpha_j$. Plugging in values of the label marginal, we can obtain $p_t(x|y = k+1)$ as $p_t(x) - \sum_{y \in \mathcal{Y}_s} p_t(y = j)p_s(x|y = j)$.

Under separability, we can obtain the label marginal $p_t$ for source classes by simply considering the set $X_{sep}$. Denote $A = [p(x|y)]_{x \in X_{sep}, y \in \mathcal{Y}_s}$ and $v = [p_t(x)]_{x \in X_{sep}}$. Then, since $A$ is full column-rank by assumption, we can define $u = (A^T A)^{-1} A^T v$. For all $x \in X_{sep}$, we have $p_t(x) =$

$\sum_{y \in \mathcal{Y}_s} p_t(y) p_s(x|y)$ and hence, $u = [p_t(y)]_{y \in \mathcal{Y}_s}$. Having obtained $[p_t(y)]_{y \in \mathcal{Y}_s}$, we recover $p_t(y = k+1) = 1 - \sum_{j \in \mathcal{Y}_s} p_t(y = j)$ and $p_t(x|y = k+1) = p_t(x) - \sum_{j \in \mathcal{Y}_s} p_t(y = j) p_s(x|y = j)$. $\quad \square$

## B.1 Examples illustrating importance of weak positivity condition

In this section, we present two examples, one, to show that weak positivity isn't sufficient for identifiability. Second, we present another example where we show that conditions in Proposition 2 are not necessary for identifiability.

**Example 1** Assume $\mathcal{X} = \{x_1, x_2, x_3, x_4, x_5\}$ and $\mathcal{Y}_t = \{1, 2, 3\}$. Suppose the $p_t(x|y = 1)$, $p_t(x|y = 2)$, and $p_t(x)$ are given as:

|       | $p_t(x|y = 1)$ | $p_t(x|y = 2)$ | $p_t(x)$ |
|-------|----------------|----------------|----------|
| $x_1$ | 0.4            | 0.56           | 0.356    |
| $x_2$ | 0.3            | 0.3            | 0.207    |
| $x_3$ | 0.2            | 0.1            | 0.09     |
| $x_4$ | 0.1            | 0.04           | 0.042    |
| $x_5$ | 0.0            | 0.0            | 0.305    |

Here, there exists two separate $p_t(x|y = 3)$ and $p_t(y)$ that are consistent with the given $p_t(x|y = 1)$, $p_t(x|y = 2)$, and $p_t(x)$ and both the solutions satisfy weak positivity for two different $X_{\text{wp}}$ and $X'_{\text{wp}}$.

In particular, notice that $p_t(x|y = 3) = [0.17, 0.0675, 0.0, 0.0, 0.7625]^T$ and $p_t(y) = [0.3, 0.3, 0.4]$ gives us the first solution. $p_t(x|y = 3) = [0.0, 0.0, 0.0645, 0.0096, 0.9839]^T$ and $p_t(y) = [0.19, 0.5, 0.31]$ gives us another solution. For solution 1, $X_{\text{wp}} = \{x_3, x_4\}$ and for solution 2, $X'_{\text{wp}} = \{x_1, x_2\}$. To check consistency of each solution notice that $\sum_{i \in \mathcal{Y}} p_t(y = i) p_t(x|y = i) = p_t(x)$ for each $x \in \mathcal{X}$. $\quad \square$

In the above example, the key is to show that absent knowledge of which $x$'s constitute the set $X_{\text{wp}}$, we might be able to obtain multiple different solutions, each with different $X_{\text{wp}}$ and both $p_t(y)$, $p_t(x|y = k+1)$ satisfying the given information and simplex constraints.

Next, we will show that in certain scenarios weak positivity is enough for identifiability.

**Example 2** Assume $\mathcal{X} = \{x_1, x_2, x_3, x_4\}$ and $\mathcal{Y}_t = \{1, 2, 3\}$. Suppose the $p_t(x|y = 1)$, $p_t(x|y = 2)$, and $p_t(x)$ are given as,

|       | $p_t(x|y = 1)$ | $p_t(x|y = 2)$ | $p_t(x)$ |
|-------|----------------|----------------|----------|
| $x_1$ | 0.5            | 0.2            | 0.24     |
| $x_2$ | 0.3            | 0.4            | 0.2      |
| $x_3$ | 0.1            | 0.35           | 0.35     |
| $x_4$ | 0.1            | 0.05           | 0.21     |

Here, out of all $^4C_2$ possibilities for $X_{\text{wp}}$, only one possibility yields a solution that satisfies weak positivity and simplex constraints. In particular, the solution is given by $p_t(x|y = 3) = [0.0, 0.0, 0.6, 0.4]^T$ and $p_t(y) = [0.4, 0.2, 0.4]$ with $X_{\text{wp}} = \{x_1, x_2\}$. $\quad \square$

In this example, we show that conditions in Proposition 2 are not necessary to ensure identifiability. For discrete domains, this example also highlights that we can check identifiability in exponential time for any OSLS problem given $p_t(x)$ and $p_s(x|y)$ for all $y \in \mathcal{Y}_s$.

## B.2 Extending identifiability conditions to continuous distributions

To extend our identifiability conditions for continuous distributions, the linear independence conditions on the matrix $[p_s(x|y)]_{x \in X_{\text{sep}}, y \in \mathcal{Y}_s}$ has the undesirable property of being sensitive to changes on sets of measure zero. In particular, by changing a collection of linearly dependent distributions on a set of measure zero, we can make them linearly independent. As a consequence, we may impose a *stronger* notion of independence, i.e., the set of distributions $\{p(x|y) : y = 1, ..., k\}$ are such

that there does not exist $v \neq 0$ for which $\int_X |\sum_y p(x|y) v_y| dx = 0$, where $X = X_{\mathrm{wp}}$ for necessary condition and $X = X_{\mathrm{sp}}$ for sufficiency. We refer this condition as *strict linear independence*.

## C  PULSE Framework

In our PULSE framework, we build on top of BBE and CVIR from Garg et al. [29]. Here, we elaborate on Step 3 and 5 in Algorithm 1.

**Extending BBE algorithm to estimate target marginal among previously seen classes**  We first explain the intuition behind BBE approach. In a PU learning problem, given positive and unlabeled data, BBE estimates the fraction of positives in unlabeled in the push-forward space of the classifier. In particular, instead of operating in the original input space, BBE maps the inputs to one-dimensional outputs (i.e., a score between zero and one) which is the predicted probability of an example being from the positive class. BBE identifies a threshold on probability scores assigned by a domain discriminator classifier such that the ratio between the fractions of positive and unlabeled points receiving scores above the threshold is minimized. Intuitively, if their exists a threshold on probability scores assigned by the classifier such that the examples mapped to a score greater than the threshold are *mostly* positive, BBE aims to identify this threshold. Efficacy of BBE procedure relies on existence of such a threshold. This is referred to as the *top bin property*. We provide empirical evidence to the property in Fig. 2 in App. D.1. We tailor BBE to estimate the relative fraction of previously seen classes in the target distribution by exploiting a $k$-way source classifier $f_s$ trained on labeled source data. We describe the procedure in Algorithm 2.

We now introduce some notation needed to introduce the tailored BBE proceudre formally. For given probability density function $p$ and a scalar output function $f$, define a function $q(z) = \int_{A_z} p(x) dx$, where $A_z = \{x \in \mathcal{X} : f(x) \geqslant z\}$ for all $z \in [0, 1]$. Intuitively, $q(z)$ captures the cumulative density of points in a top bin, the proportion of input domain that is assigned a value larger than $z$ by the function $f$ in the transformed space. We define an empirical estimator $\widehat{q}(z)$ given a set $X = \{x_1, x_2, \ldots, x_n\}$ sampled iid from $p(x)$. Let $Z = f(X)$. Define $\widehat{q}(z) = \sum_{i=1}^n \mathbb{I}[z_i \geqslant z]/n$.

Our modified BBE procedure proceeds as follows. Given a held-out dataset of source $\{\mathbf{X}_2^S, \mathbf{y}_2^S\}$ and unlabeled target samples $\mathbf{X}_2^T$, we push all examples through the source classifier $f$ to obtain $k$ dimensional outputs. For all $j \in \mathcal{Y}_s$, we repeat the following: Obtain $Z_s = f_j(\mathbf{X}_2^S[\mathrm{id}_j])$ and $Z_t = f_j(\mathbf{X}_2^T)$. Intuitively, $Z_s$ and $Z_t$ are the push forward mapping of the source classifier. Next, with $Z_p$ and $Z_u$, we estimate $\widehat{q}_s$ and $\widehat{q}_t$. Finally, we estimate $[\widehat{p}_t]_j$ as the ratio $\widehat{q}_t(\widehat{c})/\widehat{q}_s(\widehat{c})$ at $\widehat{c}$ that minimizes the upper confidence bound at a pre-specified level $\delta$ and a fixed parameter $\gamma \in (0, 1)$. Our method is summarized in Algorithm 2. Throughout all the experiments, we fix $\delta$ at 0.1 and $\gamma$ at 0.01.

---

**Algorithm 2** Extending Best Bin Estimation (BBE) for Step 3 in Algorithm 1

---

**input** : Validation source $\{\mathbf{X}_2^S, \mathbf{y}_2^S\}$ and unlabeled target samples $\mathbf{X}_2^T$. Source classifier $f : \mathcal{X} \to \Delta^{k-1}$. Hyperparameter $0 < \delta, \gamma < 1$.
1: $\widehat{p}_t \leftarrow \mathrm{zeros}(size = |\mathcal{Y}_s|)$
2: **for** $j \in \mathcal{Y}_s$ **do**
3:   $\mathrm{id}_j \leftarrow \mathrm{where}(\mathbf{y}_2^S = j)$.
4:   $Z_s, Z_t \leftarrow \left[f(\mathbf{X}_2^S[\mathrm{id}_j])\right]_j, \left[f(\mathbf{X}_2^T)\right]_j$.
5:   $\widehat{q}_s(z), \widehat{q}_t(z) \leftarrow \frac{\sum_{z_i \in Z_s} \mathbb{I}[z_i \geqslant z]}{|\mathrm{id}_j|}, \frac{\sum_{z_i \in Z_t} \mathbb{I}[z_i \geqslant z]}{|\mathbf{X}_2^T|}$ for all $z \in [0, 1]$.
6:   $\widehat{c}_j \leftarrow \arg\min_{c \in [0,1]} \left( \frac{\widehat{q}_t(c)}{\widehat{q}_s(c)} + \frac{1+\gamma}{\widehat{q}_s(c)} \left( \sqrt{\frac{\log(4/\delta)}{2|\mathbf{X}_2^T|}} + \sqrt{\frac{\log(4/\delta)}{2|\mathrm{id}_j|}} \right) \right)$.
7:   $[\widehat{p}_t]_j \leftarrow \frac{\widehat{q}_t(\widehat{c}_j)}{\widehat{q}_s(\widehat{c}_j)}$.
8: **end for**
**output** : Normalized target marginal among source classes $\widehat{p}_t' \leftarrow \frac{\widehat{p}_t}{\|\widehat{p}_t\|_1}$

---

**Extending CVIR to train discriminator $f_d$ and estimate novel class prevalence**  After estimating the fraction of source classes in target (i.e., $p_t'(j) = p_t(y=j)/\sum_{k \in \mathcal{Y}_s} p_t(y=k)$ for all $j \in \mathcal{Y}_s$), we re-sample the source data according to $p_t'(y)$ to mimic samples from distribution $p_s'(x)$. Thus, obtaining

a PU learning problem instance, we resort to PU learning techniques to (i) estimate the fraction of novel class $p_t(y = k + 1)$; and (ii) learn a binary classifier $f_d(x)$ to discriminate between label shift corrected source $p'_s(x)$ and novel class $p_t(x|y = k + 1)$. Assume that sigmoid output $f_d(x)$ indicates predicted probability of an example $x$ belonging to label shift corrected source $p'_s(x)$. With $\widehat{\mathcal{L}}^+(f_\theta; X)$, we denote the loss incurred by $f_\theta$ when classifying examples from $X$ as positive, i.e., $\widehat{\mathcal{L}}^+(f_\theta; X) = \sum_{i=1}^{|X|} \frac{\ell(f_\theta(x_i), +1)}{|X|}$. Similarly, $\widehat{\mathcal{L}}^-(f_\theta; X) = \sum_{i=1}^{|X|} \frac{\ell(f_\theta(x_i), -1)}{|X|}$

Given an estimate of the fraction of novel class $\widehat{p}_t(y = k + 1)$, CVIR objective creates a provisional set of novel examples $\mathbf{X}_1^N$ by removing $(1 - \widehat{p}_t(y = k + 1))$ fraction of examples from $\mathbf{X}_1^T$ that incur highest loss when predicted as novel class on each training epoch. Next, we update our discriminator $f_d$ by minimizing loss on label shift corrected source $\widetilde{\mathbf{X}}_1^S$ and provisional novel examples $\mathbf{X}_1^N$. This step is aimed to remove any incentive to overfit to the examples from $p'_s(x)$. Consequently, we employ the iterative procedure that alternates between estimating the prevalence of novel class $\widehat{p}_t(y = k + 1)$ (with BBE) and minimizing the CVIR loss with estimated fraction of novel class. Algorithm 3 summarizes our approach which is used in Step 3 of Algorithm 1.

Note that we need to warm start with simple domain discrimination training, since in the initial stages mixture proportion estimate is often close to 1 rejecting all the unlabeled examples. In Garg et al. [29], it was shown that the procedure is not sensitive to the choice of number of warm start epochs and in a few cases with large datasets, we can even get away without warm start (i.e., $W = 0$) without hurting the performance. In our work, we notice that given an estimate $\widehat{\alpha}$ of prevalence of novel class, we can use unbiased PU error (7) on validation data as a surrogate to identify warm start epochs for domain discriminator training. In particular, we train the domain discriminator classifier for a large number of epochs, say $E(>> W)$, and then choose the discriminator, i.e., warm start epoch $W$ at which $f_d$ achieves minimum unbiased validation loss.

Finally, to obtain a $(k + 1)$-way classifier $f_t(x)$ on target we combine discriminator $f_d$ and source classifier $f_s$ with importance-reweighted label shift correction. In particular, for all $j \in \mathcal{Y}_s$, $[f_t(x)]_j = (f_d(x)) \frac{w(j) \cdot [f_s(x)]_j}{\sum_{k \in \mathcal{Y}_s} w(k) \cdot [f_s(x)]_k}$ and $[f_t(x)]_{k+1} = 1 - f_d(x)$. Similarly, to obtain target marginal $p_t$, we re-scale the label shift estimate among previously seen classes with estimate of prevalence of novel examples, i.e., for all $j \in \mathcal{Y}_s$, assign $\widehat{p}_t(y = j) = (1 - \widehat{p}_t(y = k + 1)) \cdot \widehat{p}'_t(y = j)$.

Overall, our approach proceeds as follows (Algorithm 1): First, we estimate the label shift among previously seen classes. Then we employ importance re-weighting of source data to formulate a single PU learning problem between source and target to estimate fraction of novel class $\widehat{p}_t(y = k + 1)$ and to learn a discriminator $f_d$ for the novel class. Combining discriminator and label shift corrected source classifier we get $(k + 1)$-way target classifier.

### C.1 PULSE under separability

Our ideas for PULSE framework can be extended to separability condition since (3) continues to hold. In particular, when OSLS satisfies the separability assumption, we may hope to jointly estimate the label shift among previously seen classes with label shift estimation techniques [45, 1] and learn a domain discriminator classifier. This may be achieved by estimating label shift among examples rejected by domain discriminator classifier as belonging to previously seen classes. However, in our initial experiments, we observe that techniques proposed under strong positivity were empirically stable and outperform methods developed under separability. This is intuitive for many benchmark datasets where it may be more natural to expect that for each class there exists a subdomain that only belongs to that class than assuming separability only between novel class samples and examples from source classes.

## D    Proofs for analysis of OSLS framework

In this section, we provide missing formal statements and proofs for theorems in Sec. 8. This mainly includes analysing key steps of our PULSE procedure for target label marginal estimation (Step 3, 5 Algorithm 1) and learning the domain discriminator classifier (Step 5, Algorithm 1).

---

**Algorithm 3** Alternating between CVIR and BBE for Step 5 in Algorithm 1

---

**input** : Re-sampled training source data $\widetilde{\mathbf{X}}_1^S$, validation source data $\widetilde{\mathbf{X}}_2^S$. Training target data $\mathbf{X}_1^T$ and validation data $\mathbf{X}_2^T$. Hyperparameter $W, B, \delta, \gamma$.

1: Initialize a training model $f_\theta$ and an stochastic optimization algorithm $\mathcal{A}$.
2: $\mathbf{X}_1^N \leftarrow \mathbf{X}_1^T$.
  {// Warm start with domain discrimination training}
3: **for** $i \leftarrow 1$ to $W$ **do**
4:   Shuffle $(\widetilde{\mathbf{X}}_1^S, \mathbf{X}_1^N)$ into $B$ mini-batches. With $(\widetilde{\mathbf{X}}_1^S[i], \mathbf{X}_1^N[i])$ we denote $i^{\text{th}}$ mini-batch.
5:   **for** $i \leftarrow 1$ to $B$ **do**
6:     Set the gradient $\nabla_\theta \left[ \widehat{\mathcal{L}}^+(f_\theta; \widetilde{\mathbf{X}}_1^S[i]) + \widehat{\mathcal{L}}^-(f_\theta; \mathbf{X}_1^N[i]) \right]$ and update $\theta$ with algorithm $\mathcal{A}$.
7:   **end for**
8: **end for**
9: $\widehat{\alpha} \leftarrow \text{BBE}(\widetilde{\mathbf{X}}_2^S, \mathbf{X}_2^T, f_\theta)$                  {Algorithm 4}
10: Rank samples $x \in \mathbf{X}_1^T$ according to their loss values $\ell(f_\theta(x), -1)$.
11: $\mathbf{X}_1^N \leftarrow \{\mathbf{X}_1^T\}_{1-\widehat{\alpha}}$ where $\{\mathbf{X}_1^T\}_{1-\widehat{\alpha}}$ denote the lowest ranked $1 - \widehat{\alpha}$ fraction of samples.
12: **while** training error $\widehat{\mathcal{E}}^+(f_\theta; \widetilde{\mathbf{X}}_2^S) + \widehat{\mathcal{E}}^-(f_\theta; \mathbf{X}_1^N)$ is not converged **do**
13:   Train model $f_\theta$ for one epoch on $(\widetilde{\mathbf{X}}_1^S, \mathbf{X}_1^N)$ as in Lines 4-7.
14:   $\widehat{\alpha} \leftarrow \text{BBE}(\widetilde{\mathbf{X}}_2^S, \mathbf{X}_2^T, f_\theta)$                  {Algorithm 4}
15:   Rank samples $x \in \mathbf{X}_1^T$ according to their loss values $\ell(f_\theta(x), -1)$.
16:   $\mathbf{X}_1^N \leftarrow \{\mathbf{X}_1^T\}_{1-\widehat{\alpha}}$ where $\{\mathbf{X}_1^T\}_{1-\widehat{\alpha}}$ denote the lowest ranked $1 - \widehat{\alpha}$ fraction of samples.
17: **end while**
**output** : Trained discriminator $f_d \leftarrow f_\theta$ and novel class fraction $\widehat{p}_t(y = k + 1) \leftarrow 1 - \widehat{\alpha}$.

---

---

**Algorithm 4** Best Bin Estimation (BBE)

---

**input** : Re-sampled source data $\widetilde{\mathbf{X}}^S$ and target samples $\mathbf{X}^T$. Discriminator classifier $\widehat{f} : \mathcal{X} \to [0, 1]$. Hyperparameter $0 < \delta, \gamma < 1$.

1: $Z_s, Z_t \leftarrow f(\widetilde{\mathbf{X}}^S), f(\mathbf{X}^T)$.
2: $\widehat{q}_t(z), \widehat{q}_s(z) \leftarrow \frac{\sum_{z_i \in Z_s} \mathbb{I}[z_i \geqslant z]}{|\widetilde{\mathbf{X}}^S|}, \frac{\sum_{z_i \in Z_t} \mathbb{I}[z_i \geqslant z]}{|\mathbf{X}|^T}$ for all $z \in [0, 1]$.
3: Estimate $\widehat{c} \leftarrow \arg\min_{c \in [0,1]} \left( \frac{\widehat{q}_t(c)}{\widehat{q}_s(c)} + \frac{1+\gamma}{\widehat{q}_s(c)} \left( \sqrt{\frac{\log(4/\delta)}{2|\widetilde{\mathbf{X}}^S|}} + \sqrt{\frac{\log(4/\delta)}{2|\mathbf{X}^T|}} \right) \right)$.

**output** : $\widehat{\alpha} \leftarrow \frac{\widehat{q}_t(\widehat{c})}{\widehat{q}_s(\widehat{c})}$

---

### D.1 Formal statement and proof of Theorem 1

Before introducing the formal statement, we introduce some additional notation. Given probability density function $p$ and a source classifier $f : \mathcal{X} \to \Delta^{k-1}$, define a function $q(z, j) = \int_{A(z,j)} p(x) dx$, where $A(z, j) = \{x \in \mathcal{X} : [f(x)]_j \geqslant z\}$ for all $z \in [0, 1]$. Intuitively, $q(z, j)$ captures the cumulative density of points in a top bin for class $j$, i.e., the proportion of input domain that is assigned a value larger than $z$ by the function $f$ at the index $j$ in the transformed space. We define an empirical estimator $\widehat{q}(z, j)$ given a set $X = \{x_1, x_2, \ldots, x_n\}$ sampled iid from $p(x)$. Let $Z = [f(X)]_j$. Define $\widehat{q}(z, j) = \sum_{i=1}^n \mathbb{I}[z_i \geqslant z] / n$.

For each pdf $p_s$ and $p_t$, we define $q_s$ and $q_t$ respectively. Moreover, for each class $j \in \mathcal{Y}_s$, we define $q_{t,j}$ corresponding to $p_{t,j} := p_t(x|y = j)$ and $q_{t,-j}$ corresponding to $p_{t,-j} := \frac{\sum_{i \in \mathcal{Y}_t \setminus \{j\}} p_t(y=i) p_t(x|y=i)}{\sum_{i \in \mathcal{Y}_t \setminus \{j\}} p_t(y=j)}$. Assume that we have $n$ source examples and $m$ target examples. Now building on BBE results from Garg et al. [29], we present finite sample results for target label marginal estimation:

**Theorem 3** (Formal statement of Theorem 1). *Define* $c_j^* = \arg\min_{c \in [0,1]} (q_{t,-j}(c, j) / q_{t,j}(c, j))$, *for all* $j \in \mathcal{Y}_s$. *Assume* $\min(n, m) \geqslant \max_{j \in \mathcal{Y}_s} \left( \frac{2 \log(4k/\delta)}{q_{t,j}^2(c_j^*, j)} \right)$. *Then, for every* $\delta > 0$, $\widehat{p}_t$ *(in Algorithm 2*

*with δ as δ/k) satisfies with probability at least $1 - \delta$, we have:*

$$\|\widehat{p}_t - p_t\|_1 \leqslant \sum_{j \in \mathcal{Y}_s} (1 - p_t(y = j)) \left( \frac{q_{t,-j}(c_j^*, j)}{q_{t,j}(c_j^*, j)} \right) + \mathcal{O}\left( \sqrt{\frac{k^3 \log(4k/\delta)}{n}} + \sqrt{\frac{k^2 \log(4k/\delta)}{m}} \right).$$

When the data satisfies strong positivity, we observe that source classifiers often exhibit a threshold $c_y$ on softmax output of each class $y \in \mathcal{Y}_s$ above which the *top bin* (i.e., $[c_y, 1]$) contains mostly examples from that class $y$. Formally, as long as there exist a threshold $c_j^* \in (0, 1)$ such that $q_{t,j}(c_j^*) \geqslant \epsilon$ and $q_{t,-j}(c_j^*) = 0$ for some constant $\epsilon > 0$ for all $j \in \mathcal{Y}_s$, we show that our estimator $\widehat{\alpha}$ converges to the true $\alpha$ with convergence rate $\min(n, m)^{-1/2}$. The proof technique simply builds on the proof of Theorem 1 in Garg et al. [29]. First, we state Lemma 1 from Garg et al. [29]. Next, for completeness we provide the proof for Theorem 3 which extends proof of Theorem 1 [29] for $k$ classes.

**Lemma 1.** *Assume two distributions $q_p$ and $q_u$ with their empirical estimators denoted by $\widehat{q}_p$ and $\widehat{q}_u$ respectively. Then for every $\delta > 0$, with probability at least $1 - \delta$, we have for all $c \in [0, 1]$*

$$\left| \frac{\widehat{q}_u(c)}{\widehat{q}_p(c)} - \frac{q_u(c)}{q_p(c)} \right| \leqslant \frac{1}{\widehat{q}_p(c)} \left( \sqrt{\frac{\log(4/\delta)}{2n_u}} + \frac{q_u(c)}{q_p(c)} \sqrt{\frac{\log(4/\delta)}{2n_p}} \right).$$

*Proof of Theorem 3.* The main idea of the proof is to use the confidence bound derived in Lemma 1 at $\widehat{c}$ and use the fact that $\widehat{c}$ minimizes the upper confidence bound. The proof is split into two parts. First, we derive a lower bound on $\widehat{q}_{t,j}(\widehat{c}_j)$ for all $j \in \mathcal{Y}_s$ and next, we use the obtained lower bound to derive confidence bound on $\widehat{p}_t(y = j)$. With $\widehat{\alpha}_j$, we denote $\widehat{p}_t(y = j)$ for all $j \in \mathcal{Y}_s$. All the statements in the proof simultaneously hold with probability $1 - \delta/k$. We derive the bounds for a single $j \in \mathcal{Y}_s$ and then use union bound to combine bound for all $j \in \mathcal{Y}_s$. When it is clearly from context, we denote $q_{t,j}(c, j)$ with $q_{t,j}(c)$ and $q_t(c, j)$ with $q_t(c)$. Recall,

$$\widehat{c}_j := \arg\min_{c \in [0,1]} \frac{\widehat{q}_t(c)}{\widehat{q}_{t,j}(c)} + \frac{1}{\widehat{q}_{t,j}(c)} \left( \sqrt{\frac{\log(4k/\delta)}{2m}} + (1 + \gamma) \sqrt{\frac{\log(4k/\delta)}{2np_s(y = j)}} \right) \quad \text{and} \quad (12)$$

$$\widehat{p}_t(y = j) := \frac{\widehat{q}_t(\widehat{c}_j)}{\widehat{q}_{t,j}(\widehat{c}_j)}. \quad (13)$$

Moreover,

$$c_j^* := \arg\min_{c \in [0,1]} \frac{q_t(c)}{q_{t,j}(c)} \quad \text{and} \quad \alpha_j^* := \frac{q_t(c_j^*)}{q_{t,j}(c_j^*)}. \quad (14)$$

**Part 1:** We establish lower bound on $\widehat{q}_{t,j}(\widehat{c}_j)$. Consider $c_j' \in [0, 1]$ such that $\widehat{q}_{t,j}(c_j') = \frac{\gamma}{2+\gamma} \widehat{q}_{t,j}(c_j^*)$. We will now show that Algorithm 2 will select $\widehat{c}_j < c_j'$. For any $c \in [0, 1]$, we have with with probability $1 - \delta/k$,

$$\widehat{q}_{t,j}(c) - \sqrt{\frac{\log(4k/\delta)}{2n \cdot p_s(y = j)}} \leqslant q_{t,j}(c) \quad \text{and} \quad q_t(c) - \sqrt{\frac{\log(4k/\delta)}{2m}} \leqslant \widehat{q}_t(c). \quad (15)$$

Since $\frac{q_t(c_j^*)}{q_{t,j}(c_j^*)} \leqslant \frac{q_t(c)}{q_{t,j}(c)}$, we have

$$\widehat{q}_t(c) \geqslant q_{t,j}(c) \frac{q_t(c_j^*)}{q_{t,j}(c_j^*)} - \sqrt{\frac{\log(4k/\delta)}{2m}} \geqslant \left( \widehat{q}_{t,j}(c) - \sqrt{\frac{\log(4k/\delta)}{2n \cdot p_s(y = j)}} \right) \frac{q_t(c_j^*)}{q_{t,j}(c_j^*)} - \sqrt{\frac{\log(4k/\delta)}{2m}}. \quad (16)$$

Therefore, at $c$ we have

$$\frac{\widehat{q}_t(c)}{\widehat{q}_{t,j}(c)} \geqslant \alpha_j^* - \frac{1}{\widehat{q}_{t,j}(c)} \left( \sqrt{\frac{\log(4k/\delta)}{2m}} + \frac{q_t(c_j^*)}{q_p(c_j^*)} \sqrt{\frac{\log(4k/\delta)}{2n \cdot p_s(y = j)}} \right). \quad (17)$$

Using Lemma 1 at $c^*$, we have

$$\frac{\widehat{q}_t(c)}{\widehat{q}_{t,j}(c)} \geqslant \frac{\widehat{q}_t(c_j^*)}{\widehat{q}_{t,j}(c_j^*)} - \left(\frac{1}{\widehat{q}_{t,j}(c_j^*)} + \frac{1}{\widehat{q}_{t,j}(c)}\right)\left(\sqrt{\frac{\log(4k/\delta)}{2m}} + \frac{q_t(c_j^*)}{q_{t,j}(c_j^*)}\sqrt{\frac{\log(4k/\delta)}{2n \cdot p_s(y=j)}}\right) \quad (18)$$

$$\geqslant \frac{\widehat{q}_t(c_j^*)}{\widehat{q}_{t,j}(c_j^*)} - \left(\frac{1}{\widehat{q}_{t,j}(c_j^*)} + \frac{1}{\widehat{q}_{t,j}(c)}\right)\left(\sqrt{\frac{\log(4k/\delta)}{2m}} + \sqrt{\frac{\log(4k/\delta)}{2n \cdot p_s(y=j)}}\right), \quad (19)$$

where the last inequality follows from the fact that $\alpha_j^* = \frac{q_t(c_j^*)}{q_{t,j}(c_j^*)} \leqslant 1$. Furthermore, the upper confidence bound at $c$ is lower bound as follows:

$$\frac{\widehat{q}_t(c)}{\widehat{q}_{t,j}(c)} + \frac{1+\gamma}{\widehat{q}_{t,j}(c)}\left(\sqrt{\frac{\log(4l/\delta)}{2m}} + \sqrt{\frac{\log(4k/\delta)}{2n \cdot p_s(y=j)}}\right) \quad (20)$$

$$\geqslant \frac{\widehat{q}_t(c_j^*)}{\widehat{q}_{t,j}(c_j^*)} + \left(\frac{1+\gamma}{\widehat{q}_{t,j}(c)} - \frac{1}{\widehat{q}_{t,j}(c_j^*)} - \frac{1}{\widehat{q}_{t,j}(c)}\right)\left(\sqrt{\frac{\log(4k/\delta)}{2m}} + \sqrt{\frac{\log(4k/\delta)}{2n \cdot p_s(y=j)}}\right) \quad (21)$$

$$= \frac{\widehat{q}_t(c_j^*)}{\widehat{q}_{t,j}(c_j^*)} + \left(\frac{\gamma}{\widehat{q}_{t,j}(c)} - \frac{1}{\widehat{q}_{t,j}(c_j^*)}\right)\left(\sqrt{\frac{\log(4k/\delta)}{2m}} + \sqrt{\frac{\log(4k/\delta)}{2n \cdot p_s(y=j)}}\right) \quad (22)$$

Using (22) at $c = c'$, we have the following lower bound on ucb at $c'$:

$$\frac{\widehat{q}_t(c')}{\widehat{q}_{t,j}(c')} + \frac{1+\gamma}{\widehat{q}_{t,j}(c')}\left(\sqrt{\frac{\log(4k/\delta)}{2m}} + \sqrt{\frac{\log(4k/\delta)}{2n \cdot p_s(y=j)}}\right) \quad (23)$$

$$\geqslant \frac{\widehat{q}_t(c_j^*)}{\widehat{q}_{t,j}(c_j^*)} + \frac{1+\gamma}{\widehat{q}_{t,j}(c_j^*)}\left(\sqrt{\frac{\log(4k/\delta)}{2m}} + \sqrt{\frac{\log(4k/\delta)}{2n \cdot p_s(y=j)}}\right), \quad (24)$$

Moreover from (22), we also have that the lower bound on ucb at $c \geqslant c'$ is strictly greater than the lower bound on ucb at $c'$. Using definition of $\widehat{c}$, we have

$$\frac{\widehat{q}_t(c_j^*)}{\widehat{q}_{t,j}(c_j^*)} + \frac{1+\gamma}{\widehat{q}_{t,j}(c_j^*)}\left(\sqrt{\frac{\log(4k/\delta)}{2m}} + \sqrt{\frac{\log(4k/\delta)}{2n \cdot p_s(y=j)}}\right) \quad (25)$$

$$\geqslant \frac{\widehat{q}_t(\widehat{c})}{\widehat{q}_{t,j}(\widehat{c})} + \frac{1+\gamma}{\widehat{q}_{t,j}(\widehat{c})}\left(\sqrt{\frac{\log(4k/\delta)}{2m}} + \sqrt{\frac{\log(4k/\delta)}{2n \cdot p_s(y=j)}}\right), \quad (26)$$

and hence

$$\widehat{c} \leqslant c'. \quad (27)$$

**Part 2:** We now establish an upper and lower bound on $\widehat{\alpha}_j$. We start with upper confidence bound on $\widehat{\alpha}_j$. By definition of $\widehat{c}_j$, we have

$$\frac{\widehat{q}_t(\widehat{c})}{\widehat{q}_{t,j}(\widehat{c})} + \frac{1+\gamma}{\widehat{q}_{t,j}(\widehat{c})}\left(\sqrt{\frac{\log(4k/\delta)}{2m}} + \sqrt{\frac{\log(4k/\delta)}{2n \cdot p_s(y=j)}}\right) \quad (28)$$

$$\leqslant \min_{c\in[0,1]}\left[\frac{\widehat{q}_t(c)}{\widehat{q}_{t,j}(c)} + \frac{1+\gamma}{\widehat{q}_{t,j}(c)}\left(\sqrt{\frac{\log(4k/\delta)}{2m}} + \sqrt{\frac{\log(4k/\delta)}{2n \cdot p_s(y=j)}}\right)\right] \quad (29)$$

$$\leqslant \frac{\widehat{q}_t(c_j^*)}{\widehat{q}_{t,j}(c_j^*)} + \frac{1+\gamma}{\widehat{q}_{t,j}(c_j^*)}\left(\sqrt{\frac{\log(4k/\delta)}{2m}} + \sqrt{\frac{\log(4k/\delta)}{2n \cdot p_s(y=j)}}\right). \quad (30)$$

Using Lemma 1 at $c_j^*$, we get

$$\frac{\widehat{q}_t(c_j^*)}{\widehat{q}_{t,j}(c_j^*)} \leqslant \frac{q_t(c_j^*)}{q_{t,j}(c_j^*)} + \frac{1}{\widehat{q}_{t,j}(c_j^*)} \left( \sqrt{\frac{\log(4k/\delta)}{2m}} + \frac{q_t(c_j^*)}{q_{t,j}(c_j^*)} \sqrt{\frac{\log(4k/\delta)}{2n \cdot p_s(y=j)}} \right)$$

$$= \alpha_j^* + \frac{1}{\widehat{q}_{t,j}(c_j^*)} \left( \sqrt{\frac{\log(4k/\delta)}{2m}} + \alpha_j^* \sqrt{\frac{\log(4k/\delta)}{2n \cdot p_s(y=j)}} \right) . \tag{31}$$

Combining (30) and (31), we get

$$\widehat{\alpha}_j = \frac{\widehat{q}_t(\widehat{c})}{\widehat{q}_{t,j}(\widehat{c})} \leqslant \alpha_j^* + \frac{2+\gamma}{\widehat{q}_{t,j}(c_j^*)} \left( \sqrt{\frac{\log(4k/\delta)}{2m}} + \sqrt{\frac{\log(4k/\delta)}{2n \cdot p_s(y=j)}} \right) . \tag{32}$$

Using DKW inequality on $\widehat{q}_{t,j}(c_j^*)$, we have $\widehat{q}_{t,j}(c_j^*) \geqslant q_{t,j}(c_j^*) - \sqrt{\frac{\log(4k/\delta)}{2n \cdot p_s(y=j)}}$. Assuming $n \cdot p_s(y = j) \geqslant \frac{2 \log(4k/\delta)}{q_{t,j}^2(c_j^*)}$, we get $\widehat{q}_{t,j}(c_j^*) \leqslant q_{t,j}(c_j^*)/2$ and hence,

$$\widehat{\alpha}_j \leqslant \alpha_j^* + \frac{4+2\gamma}{q_{t,j}(c_j^*)} \left( \sqrt{\frac{\log(4k/\delta)}{2m}} + \sqrt{\frac{\log(4k/\delta)}{2n \cdot p_s(y=j)}} \right) . \tag{33}$$

Finally, we now derive a lower bound on $\widehat{\alpha}_j$. From Lemma 1, we have the following inequality at $\widehat{c}$

$$\frac{q_t(\widehat{c})}{q_{t,j}(\widehat{c})} \leqslant \frac{\widehat{q}_t(\widehat{c})}{\widehat{q}_{t,j}(\widehat{c})} + \frac{1}{\widehat{q}_{t,j}(\widehat{c})} \left( \sqrt{\frac{\log(4k/\delta)}{2m}} + \frac{q_t(\widehat{c})}{q_{t,j}(\widehat{c})} \sqrt{\frac{\log(4k/\delta)}{2n \cdot p_s(y=j)}} \right) . \tag{34}$$

Since $\alpha_j^* \leqslant \frac{q_t(\widehat{c})}{q_{t,j}(\widehat{c})}$, we have

$$\alpha_j^* \leqslant \frac{q_t(\widehat{c})}{q_{t,j}(\widehat{c})} \leqslant \frac{\widehat{q}_t(\widehat{c})}{\widehat{q}_{t,j}(\widehat{c})} + \frac{1}{\widehat{q}_{t,j}(\widehat{c})} \left( \sqrt{\frac{\log(4k/\delta)}{2m}} + \frac{q_t(\widehat{c})}{q_{t,j}(\widehat{c})} \sqrt{\frac{\log(4k/\delta)}{2n \cdot p_s(y=j)}} \right) . \tag{35}$$

Using (33), we obtain a very loose upper bound on $\frac{\widehat{q}_t(\widehat{c})}{\widehat{q}_{t,j}(\widehat{c})}$. Assuming $\min(n \cdot p_s(y = j), m) \geqslant \frac{2 \log(4k/\delta)}{q_{t,j}^2(c_j^*)}$, we have $\frac{\widehat{q}_t(\widehat{c})}{\widehat{q}_{t,j}(\widehat{c})} \leqslant \alpha_j^* + 4 + 2\gamma \leqslant 5 + 2\gamma$. Using this in (35), we have

$$\alpha_j^* \leqslant \frac{\widehat{q}_t(\widehat{c})}{\widehat{q}_{t,j}(\widehat{c})} + \frac{1}{\widehat{q}_{t,j}(\widehat{c})} \left( \sqrt{\frac{\log(4k/\delta)}{2m}} + (5+2\gamma) \sqrt{\frac{\log(4k/\delta)}{2n \cdot p_s(y=j)}} \right) . \tag{36}$$

Moreover, as $\widehat{c} \geqslant c'$, we have $\widehat{q}_{t,j}(\widehat{c}) \geqslant \frac{\gamma}{2+\gamma} \widehat{q}_{t,j}(c_j^*)$ and hence,

$$\alpha_j^* - \frac{\gamma+2}{\gamma \widehat{q}_{t,j}(c_j^*)} \left( \sqrt{\frac{\log(4k/\delta)}{2m}} + (5+2\gamma) \sqrt{\frac{\log(4k/\delta)}{2n \cdot p_s(y=j)}} \right) \leqslant \frac{\widehat{q}_t(\widehat{c})}{\widehat{q}_{t,j}(\widehat{c})} = \widehat{\alpha}_j . \tag{37}$$

As we assume $n \cdot p_s(y = j) \geqslant \frac{2 \log(4k/\delta)}{q_{t,j}^2(c_j^*)}$, we have $\widehat{q}_{t,j}(c_j^*) \leqslant q_{t,j}(c_j^*)/2$, which implies the following lower bound on $\alpha$:

$$\alpha_j^* - \frac{2\gamma+4}{\gamma q_{t,j}(c_j^*)} \left( \sqrt{\frac{\log(4k/\delta)}{2m}} + (5+2\gamma) \sqrt{\frac{\log(4k/\delta)}{2n \cdot p_s(y=j)}} \right) \leqslant \widehat{\alpha}_j . \tag{38}$$

Combining lower bound (38) and upper bound (33), we get

$$\left| \widehat{\alpha}_j - \alpha_j^* \right| \leqslant l_j \left( \sqrt{\frac{\log(4k/\delta)}{2m}} + \sqrt{\frac{\log(4k/\delta)}{2n \cdot p_s(y=j)}} \right) , \tag{39}$$

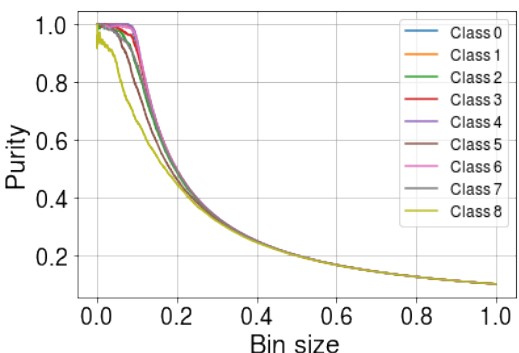

Figure 2: Purity and size (in terms of fraction of unlabeled samples) in the top bin for all classes. Bin size refers to the fraction of examples in the top bin. With purity, we refer to the fraction of examples from a specific class $j$ in the top bin. Results with ResNet-18 on CIFAR10 OSLS setup. Details of the setup in App. F.2. As the bin size increases for all classes the purity decreases.

for some constant $l_j$. Additionally by our assumption of OSLS problem $p_s(y = j) > c/k$ for some constant $c > 0$, we have

$$\left|\widehat{\alpha}_j - \alpha_j^*\right| \leqslant l'_j \left( \sqrt{\frac{\log(4k/\delta)}{2m}} + \sqrt{\frac{k\log(4k/\delta)}{2n}} \right), \tag{40}$$

for some constant $l'_j$.

Combining the above obtained bound for all $j \in \mathcal{Y}_s$ with union bound, we get with probability at least $1 - \delta$,

$$\sum_{j \in \mathcal{Y}_s} \left|\widehat{\alpha}_j - \alpha_j^*\right| \leqslant l'_{\max} \left( \sqrt{\frac{k^2 \log(4k/\delta)}{2m}} + \sqrt{\frac{k^3 \log(4k/\delta)}{2n}} \right), \tag{41}$$

where $l'_{\max} = \max l'_j$. Now, note that for each $j \in \mathcal{Y}_s$, we have $q_t(c) = p_t(y = j) \cdot q_{t,j}(c) + (1 - p_t(y = j)) \cdot q_{t,-j}(c)$. Hence $\alpha_j^* = p_t(y = j) + (1 - p_t(y = j)) \cdot q_{t,-j}(c) / \cdot q_{t,j}(c)$. Plugging this in, we get the desired bound. $\qquad\square$

Intuitively, the guarantees in the previous theorem capture the tradeoff due to the proportion of negative examples in the top bin (bias) versus the proportion of positives in the top bin (variance). As a corollary, we can show convergence to true mixture if there exits $c_j^*$ for all $j \in \mathcal{Y}_s$ such that $q_{t,-j}(c_j^*, j) = 0$ and $q_{t,j}(c_j^*, j) \geqslant \epsilon$ for some $\epsilon > 0$. Put simply, efficacy of BBE relies on existence of a threshold on probability scores assigned by the classifier such that the examples mapped to a score greater than the threshold are *mostly* positive. Using the terminology from Garg et al. [29], we refer to this as the top bin property. Next, we provide empirical evidence of this property while using the source classifier to estimate the relative proportion of target label marginal among source classes.

**Empirical evidence of the top bin property** We now empirically validate the positive pure top bin property (Fig. 2). We include results with Resnet-18 trained on the CIFAR10 OSLS setup same as our main experiments. We observe that source classifier approximately satisfies the positive pure top bin property for small enough top bin sizes.

### D.2 Formal statement and proof of Theorem 2

In this section, we show that in population on a separable Gaussian dataset, CVIR will recover the optimal classifier. Note that here we consider a binary classification problem similar to the one in Step 5 in Algorithm 1. Since we are primarily interested in analysing the iterative procedure for obtaining domain discriminator classifier, we assume that $\alpha$ is known.

In population, we have access to positive distribution (i.e., $p_p$), unlabeled distribution (i.e., $p_u := \alpha p_p + (1 - \alpha)p_n$), and mixture coefficient $\alpha$. Our goal is to recover the classifier that discriminates $p_p$ versus $p_n$.

For ease, we re-introduce some notation. For a classifier $f$ and loss function $\ell$, define
$$\text{VIR}_\alpha(f) = \inf\{\tau \in \mathbb{R} : \text{P}_{x \sim p_u}(\ell(x, -1; f) \leqslant \tau) \geqslant 1 - \alpha\}. \tag{42}$$
Intuitively, $\text{VIR}_\alpha(f)$ identifies a threshold $\tau$ to capture bottom $1 - \alpha$ fraction of the loss $\ell(x, -1)$ for points $x$ sampled from $p_u$. Additionally, define CVIR loss as
$$\mathcal{L}(f, w) = \alpha \mathbb{E}_{p_p}\left[\ell(x, 1; f)\right] + \mathbb{E}_{p_u}\left[w(x)\ell(x, -1; f)\right], \tag{43}$$
for classifier $f$ and some weights $w(x) \in \{0, 1\}$. Recall that given a classifier $f_t$ at an iterate $t$, CVIR procedure proceeds as follows:
$$w_t(x) = \mathbb{I}\left[\ell(x, -1; f_t) \leqslant \text{VIR}_\alpha(f_t)\right], \tag{44}$$
$$f_{t+1} = f_t - \eta \nabla \mathcal{L}_f(f_t, w_t). \tag{45}$$

We assume a data generating setup with where the support of positive and negative data is completely disjoint. We assume that $x$ are drawn from two half multivariate Gaussian with mean zero and identity covariance, i.e.,
$$x \sim p_p \Leftrightarrow x = \gamma_0 \theta_{\text{opt}} + z \,|\, \theta_{\text{opt}}^T z \geqslant 0, \text{ where } z \sim \mathcal{N}(0, I_d)$$
$$x \sim p_n \Leftrightarrow x = -\gamma_0 \theta_{\text{opt}} + z \,|\, \theta_{\text{opt}}^T z < 0, \text{ where } z \sim \mathcal{N}(0, I_d)$$

Here $\gamma_0$ is the margin and $\theta_{\text{opt}} \in \mathbb{R}^d$ is the true separator. Here, we have access to distribution $p_p$ and $p_u = \alpha p_p + (1 - \alpha)p_n$. Assume $\ell$ as the logistic loss. For simplicity, we will denote $\mathcal{L}(f_{\theta_t}, w_t)$ with $\mathcal{L}(\theta_t, w_t)$.

**Theorem 4** (Formal statement of Theorem 2). *In the data setup described above, a linear classifier $f(x; \theta) = \sigma\left(\theta^T x\right)$ initialized at some $\theta_0$ such that $\mathcal{L}(\theta_0, w_0) < \log(2)$, trained with CVIR procedure as in equations (44)-(45) will converge to an optimal positive versus negative classifier.*

*Proof of Theorem 4.* The proof uses two key ideas. One, at convergence of the CVIR procedure, the gradient of CVIR loss in (43) converges to zero. Second, for any classifier $\theta$ that is not optimal for positive versus negative classification, we show that the CVIR gradient in (43) is non-zero.

**Part 1** We first show that the loss function $\mathcal{L}(\theta, w)$ in (43) is 2-smooth with respect to $\theta$ for fixed $w$. Using gradient descent lemma with the decreasing property of loss in (44)-(45), we show that gradient converges to zero eventually. Considering gradient of $\mathcal{L}$, we have
$$\nabla_\theta \mathcal{L}(\theta, w) = \alpha \mathbb{E}_{p_p}\left[(f(x; \theta) - 1)x\right] + \mathbb{E}_{p_u}\left[w(x)(f(x; \theta) - 0)x\right]. \tag{46}$$
Moreover, $\nabla^2 \mathcal{L}$ is given by
$$\nabla_\theta^2 \mathcal{L}(\theta, w) = \alpha \mathbb{E}_{p_p}\left[\nabla f(x; \theta) x x^T\right] + \mathbb{E}_{p_u}\left[w(x)\nabla f(x; \theta) x x^T\right]. \tag{47}$$
Since $\nabla f(x; \theta) \leqslant 1$, we have $v^T \nabla^2 \mathcal{L} v \leqslant 2$ for all unit vector $v \in R^d$. Now, by gradient descent lemma if $\eta \leqslant 1/2$, at any step $t$ we have, $\mathcal{L}(\theta_{t+1}, w_t) \leqslant \mathcal{L}(\theta_t, w_t)$. Moreover, by definition of $\text{VIR}_\alpha(\theta)$ in (42) and update (44), we have $\mathcal{L}(\theta_{t+1}, w_{t+1}) \leqslant \mathcal{L}(\theta_{t+1}, w_t)$. Hence, we have $\mathcal{L}(\theta_{t+1}, w_{t+1}) \leqslant \mathcal{L}(\theta_t, w_t)$. Since, the loss is lower bounded from below at 0, for every $\epsilon > 0$, we have for large enough $t$ (depending on $\epsilon$), $\|\nabla_\theta \mathcal{L}(\theta_t, w_t)\|_2 \leqslant \epsilon$, i.e., $\|\nabla_\theta \mathcal{L}(\theta_t, w_t)\|_2 \to 0$ as $t \to \infty$.

**Part 2** Consider a general scenario when $\gamma > 0$. Denote the input domain of $p_p$ and $p_n$ as $P$ and $N$ respectively. At any step $t$, for all points $x \in \mathcal{X}$ such that $p_u(x) > 0$ and $w_t(x) = 0$, we say that $x$ is rejected from $p_u$. We denote the incorrectly rejected subdomain of $p_n$ from $p_u$ as $N_r$ and the incorrectly accepted subdomain of $p_p$ from $p_u$ as $P_a$. Formally, $N_r = \{x : p_n(x) > 0 \text{ and } w_t(x) = 0\}$ and $P_a = \{x : p_p(x) > 0 \text{ and } w_t(x) = 1\}$. We will show that $p_p(P_a) \to 0$ as $t \to \infty$, and hence, we will recover the optimal classifier where we reject none of $p_u$ incorrectly.

Observe that at any time $t$, for fixed $w_t$ and $\theta = \theta_t$, the gradient of CVIR loss in (43), can be expressed as:
$$\nabla_\theta \mathcal{L}(\theta, w_t) = \alpha \underbrace{\int_{x \in P \backslash P_a} (f(x; \theta) - 1)x \cdot p_p(x)dx}_{\text{I}} + (1 - \alpha)\underbrace{\int_{x \in N \backslash N_r} (f(x; \theta) - 0)x \cdot p_n(x)dx}_{\text{II}}$$
$$+ \alpha \underbrace{\int_{x \in P_a} (2f(x; \theta) - 1)x \cdot p_p(x)dx}_{\text{III}}. \tag{48}$$

Note that for any $x, \theta, 0 \leqslant f(x; \theta) \leqslant 1$. Now consider inner product of individual terms above with $\theta_{\text{opt}}$, we get

$$\langle \text{I}, \theta_{\text{opt}} \rangle = \int_{x \in P \backslash P_a} (f(x; \theta) - 1) x^T \theta_{\text{opt}} \cdot p_p(x) dx \leqslant -\gamma_0 \int_{x \in P \backslash P_a} (1 - f(x; \theta)) \cdot p_p(x) dx, \quad (49)$$

$$\langle \text{II}, \theta_{\text{opt}} \rangle = \int_{x \in N \backslash N_r} (f(x; \theta) - 0) x^T \theta_{\text{opt}} \cdot p_n(x) dx \leqslant -\gamma_0 \int_{x \in N \backslash N_r} (f(x; \theta) - 0) \cdot p_n(x) dx, \quad (50)$$

$$\langle \text{III}, \theta_{\text{opt}} \rangle = \int_{x \in P_a} (2f(x; \theta) - 1) x^T \theta_{\text{opt}} \cdot p_p(x) dx \leqslant -\gamma_0 \int_{x \in P_a} (1 - 2f(x; \theta)) \cdot p_p(x) dx. \quad (51)$$

Now, we will argue that individually all the three LHS terms in (49), (50), (51) are negative for all classifiers that do not separate positive versus negative data begining from $\mathcal{L}(\theta_0, w_0) < \log(2)$. And hence, we show that these terms approach zero individually only when the linear classifier approaches an optimal positive versus negative classifier.

First, we consider the term in the LHS of equation (51). When $\alpha = 0.5$, we have $\text{VIR}_\alpha(\theta) = 0.5$ and hence, $(1 - 2f(x; \theta)) \leqslant 0$ for $x \in P_a$. When $\alpha > 0.5$, $\text{VIR}_\alpha(\theta) < 0.5$ because, the proportion $\alpha \cdot p_p(P_a)$ matches with proportion $(1 - \alpha) \cdot p_n(N_r)$. Hence, we again have $(1 - 2f(x; \theta)) \leqslant 0$ for $x \in P_a$.

To handle the case with $\alpha < 0.5$, we use a symmetry of he distribution to because $\text{VIR}_\alpha(\theta) > 0.5$ and $(1 - 2f(x; \theta))$ can take positive and negative values. However, note that $\text{VIR}_\alpha(\theta)$ will be selected such that the proportion $\alpha \cdot p_p(P_a)$ matches with proportion $(1 - \alpha) \cdot P_n(N_r)$. In particular, we can split $P_a$ into three disjoint sets $P_a^{(1)}$, $P_a^{(2)}$, and $P_a^{(3)}$ such that for all $x \in P_a^{(1)}$ we have $f(x; \theta) >= 0.5$, for all $x \in P_a^{(2)} \cup P_a^{(3)}$ we have $f(x; \theta) < 0.5$ and $p_p(P_a^{(3)}) = \frac{\alpha}{1-\alpha} p_p(N_r)$. Additionally, by symmetry of distribution around $\theta$, we have $\int_{x \in P_a^{(1)}} (1 - 2f(x; \theta)) \cdot p_p(x) dx + \int_{x \in P_a^{(2)}} (1 - 2f(x; \theta)) \cdot p_p(x) dx = 0$. Hence, we get

$$\langle \text{III}, \theta_{\text{opt}} \rangle \leqslant -\gamma_0 \int_{x \in P_a} (1 - 2f(x; \theta)) \cdot p_p(x) dx = -\gamma_0 \int_{x \in P_a^{(3)}} (1 - 2f(x; \theta)) \cdot p_p(x) dx. \quad (52)$$

Combining all three cases, we get $\langle \text{III}, \theta_{\text{opt}} \rangle < 0$ when $p_p(P_a) > 0$.

Now we consider LHS terms in (49) and (50). Note that for all $x \in P \cup N$, we have $0 \leqslant f(x) \leqslant 1$. Thus with $p_p(P \backslash P_a) > 0$, $\langle \text{I}, \theta_{\text{opt}} \rangle \to 0$ when $f(x, \theta) \to 1$ for all $x \in P \backslash P_a$. Similarly with $p_n(N \backslash N_r) > 0$, $\langle \text{II}, \theta_{\text{opt}} \rangle \to 0$ when $f(x, \theta) \to 0$ for all $x \in N \backslash N_r$.

From part 1, for gradient $\|\nabla_\theta \mathcal{L}(\theta_t, w_t)\|_2$ to converge to zero as $t \to \infty$, we must have that LHS in equations (49), (50), and (51) converges to zero individually. Since CVIR loss decreases continuously and $\mathcal{L}(\theta_0, w_0) < \log(2)$, we have that $p_p(P_a) \to 0$ and hence, $f(x, \theta) \to 1$ for all $x \in P$ and $f(x, \theta) \to 0$ for all $x \in N$.

$\square$

The above analysis can be extended to show convergence to max-margin classifier by using arguments from Soudry et al. [65]. In particular, as $p_p(P_a) \to 0$, we can show that $\theta_t / \|\theta_t\|_2$ will converge to the max-margin classifier for $p_p$ versus $p_n$, i.e., $\theta_{\text{opt}}$ if $p_p(P_a) \to 0$ in finite number of steps. Note that we need an assumption that the initialized model $\theta_0$ is strictly better than a model that randomly guesses or initialized at all zeros. This is to avoid convergence to the local minima of $\theta = \mathbf{0}$ with CVIR training. This assumption is satisfied when the classifier is initialized in a way such that $\langle \theta_0, \theta_{\text{opt}} \rangle > 0$. In general, we need a weaker assumption that during training with any randomly initialized classifier, there exists an iterate $t$ during CVIR training such that $\langle \theta_t, \theta_{\text{opt}} \rangle > 0$.

### D.3    Extension of Theorem 1

We also extend the analysis in the proof of Theorem 3 to Step 5 of Algorithm 1 to show convergence of estimate $\hat{p}_t(y = k + 1)$ to true prevalence $p_t(y = k + 1)$. In particular, we show that the estimation error for prevalence of the novel class will primarily depend on sum of two terms: (i) error in

approximating the label shift corrected source distribution, i.e., $p'_s(x)$; and (ii) purity of the top bin of the domain discriminator classifier.

Before formally introducing the result, we introduce some notation. Similar to before, given probability density function $p$ and a domain discriminator classifier $f : \mathcal{X} \to \Delta$, define a function $q = \int_{A(z)} p(x)dx$, where $A(z) = \{x \in \mathcal{X} : f(x) \geqslant z\}$ for all $z \in [0,1]$. Intuitively, $q(z)$ captures the cumulative density of points in a top bin, i.e., the proportion of input domain that is assigned a value larger than $z$ by the function $f$ in the transformed space. We denote $p_t(x|y = k+1)$ with $p_{t,k+1}$. For each pdf $p_t$, $p_{t,k+1}$, and $p'_s$, we define $q_t$, $q_{t,k+1}$, and $q'_s$ respectively. Note that since We define an empirical estimator $\widehat{q}(z)$ given a set $X = \{x_1, x_2, \ldots, x_n\}$ sampled iid from $p(x)$. Let $Z = f(X)$. Define $\widehat{q}(z) = \sum_{i=1}^{n} \mathbb{I}[z_i \geqslant z]/n$.

Recall that in Step 5 of Algorithm 1, to estimate the proportion of novel class, we have access to re-sampled data from approximate label shift corrected source distribution $\widehat{q}'_s(x)$. Assume that we the size of re-sampled dataset is $n$.

**Theorem 5.** *Define $c^* = \arg\min_{c \in [0,1]} (q_{t,k+1}(c)/\widehat{q}'_s(c))$. Assume $\min(n,m) \geqslant \left( \frac{2\log(4/\delta)}{(\widehat{q}'_s(c^*))^2} \right)$. Then, for every $\delta > 0$, $[\widehat{p}_t]_{k+1} := \widehat{p}_t(y = k+1)$ in Step 5 of Algorithm 1 satisfies with probability at least $1 - \delta$, we have:*

$$|[\widehat{p}_t]_{k+1} - [p_t]_{k+1}| \leqslant (1 - [p_t]_{k+1}) \underbrace{\frac{|q'_s(c^*) - \widehat{q}'_s(c^*)|}{\widehat{q}'_s(c^*)}}_{\substack{\text{Error in estimating} \\ \text{label shift corrected source}}} + [p_t]_{k+1} \underbrace{\left( \frac{q_{t,k+1}(c^*)}{\widehat{q}'_s(c^*)} \right)}_{\substack{\text{Impurity in} \\ \text{top bin}}}$$

$$+ \mathcal{O}\left( \sqrt{\frac{\log(4/\delta)}{n}} + \sqrt{\frac{\log(4/\delta)}{m}} \right) .$$

*Proof.* We can simply prove this theorem as Corollary of Theorem 1 from Garg et al. [29]. Note that $q_t(c^*) = (1 - p_t(y = k+1)) \cdot q'_s(c^*) + p_t(y = k+1) \cdot q_{t,k+1}(c^*)$. Adding and subtracting $(1 - p_t(y = k+1)) \cdot \widehat{q}'_s(c^*)$ and dividing by $\widehat{q}'_s$, we get $\frac{q_t(c^*)}{\widehat{q}'_s(c^*)} = (1 - p_t(y = k+1)) \cdot \frac{|q'_s(c^*) - \widehat{q}'_s(c^*)|}{\widehat{q}'_s(c^*)} + (1 - p_t(y = k+1)) + p_t(y = k+1) \cdot \frac{q_{t,k+1}(c^*)}{\widehat{q}'_s(c^*)}$. Plugging in bound for LHS from Theorem 1 in Garg et al. [29], we get the desired result. $\square$

### D.4 Extensions of Theorem 2 to general separable datasets

For general separable datasets, CVIR has undesirable property of getting stuck at local optima where gradient in (51) can be zero by maximizing entropy on the subset $P_a$ which is (incorrectly) not-rejected from $p_u$ in CVIR iterations. Intuitively, if the classifier can perfectly separate $P \backslash P_a$ and $N \backslash N_r$ and at the same time maximize the entropy of the region $P_a$, then the classifier trained with CVIR can get stuck in this local minima.

However, we can extend the above analysis with some modifications to the CVIR procedure. Note that when the CVIR classifier maximizes the entropy on $P_a$. it makes an error on points in $P_a$. Since, we have access to the distribution $p_p$, we can add an additional regularization penalty to the CVIR loss that ensures that the converged classifier with CVIR correctly classifies all the points in $p_p$. With a large enough regularization constant for the supervised loss on $p_p$, we can dominate the gradient term in (51) which pushes CVIR classifier to correct decision boundary even on $P_a$ (instead of maximizing entropy). We leave formal analysis of this conjecture for future work. Since we warm start CVIR training with a positive versus unlabeled classifier, if we obtain an initialization close enough to the true positive versus negative decision boundary, by monotonicity property of CVIR iterations, we may never get stuck in such a local minima even without modifications to loss.

## E    Empirical investigation of CVIR in toy setup

As noted in our ablation experiments and in Garg et al. [29], domain discriminator trained with CVIR outperforms classifiers trained with other consistent objectives (nnPU [38] and uPU [21]). While the analysis in Sec. 8 highlights consistency of CVIR procedure in population, it doesn't capture the

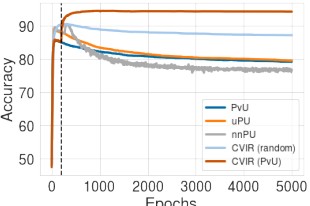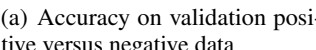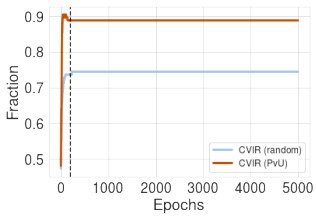

(a) Accuracy on validation positive versus negative data

(b) Fraction of correctly rejected examples with CVIR

Figure 3: **Comparison of different methods in overparameterized toy setup.** CVIR (random) denotes CVIR with random initialization and CVIR (PvU) denotes warm start with a positive versus negative classifier. Vertical line denotes the epoch at which we switch from PvU to CVIR in CVIR (PvU) training. (a) We observe that CVIR (PvU) improves significantly even over the best early stopped PvU model. As training proceeds, we observe that accuracy of nnPU, uPU and PvU training drops whereas CVIR (random) and CVIR (PvU) maintains superior and stable performance. (b) We observe that warm start training helps CVIR over randomly initialized model to correctly identity positives among unlabeled for rejection.

observed empirical efficacy of CVIR over alternative methods in overparameterized models. In the Gaussian setup described in Sec. D.2, we train overparameterized linear models to compare CVIR with other methods (Fig. 3). We fix $d = 1000$ and use $n = 250$ positive and $m = 250$ unlabeled points for training with $\alpha = 0.5$. We set the margin $\gamma$ at $0.05$. We compare CVIR with unbiased losses uPU and nnPU. We also make comparison with a naive positive versus unlabeled classifier (referred to as PvU). For CVIR, we experiment with a randomly initialized classifier and initialized with a PvU classifier trained for 200 epochs.

First, we observe that when a classifier is trained to distinguish positive and unlabeled data, *early learning* happens [47, 3, 28], i.e., during the initial phase of learning classifier learns to classify positives in unlabeled correctly as positives achieving high accuracy on validation positive versus negative data. While the early learning happens with all methods, soon in the later phases of training PvU starts overfitting to the unlabeled data as negative hurting its validation performance. For uPU and nnPU, while they improve over PvU training during the initial epochs, the loss soon becomes biased hurting the performance of classifiers trained with uPU and nnPU on validation data.

For CVIR trained from a randomly initialized classifier, we observe that it improves slightly over the best PvU or the best nnPU model. Moreover, it maintains a relatively stable performance throughout the training. CVIR initialized with a PvU classifier significantly improves the performance. In Fig. 3 (b), we show that CVIR initialized with a PvU correctly rejects significantly more fraction of positives from unlabeled than CVIR trained from scratch. Thus, post early learning rejection of large fraction of positives from unlabeled training in equation (4) crucially helps CVIR.

# F   Experimental Details

## F.1   Baselines

We compare PULSE with several popular methods from OSDA literature. While these methods are not specifically proposed for OSLS, they are introduced for the more general OSDA problem. In particular, we make comparions with DANCE [59], UAN [73], CMU [25], STA [46], Backprop-ODA (or BODA) [58]. We use the open source implementation available at `https://github.com/thuml` and `https://github.com/VisionLearningGroup/DANCE/`. Since OSDA methods do not estimate the prevalence of novel class explicitly, we use the fraction of examples predicted in class $k + 1$ as a surrogate. We next briefly describe the main idea for each method:

*Backprob-ODA*   Saito et al. [58] proposed backprob ODA to train a $(k + 1)$-way classifier. In particular, the network is trained to correctly classify source samples and for target samples, the classifier (specifically the last layer) is trained to output $0.5$ for the probability of the unknown class.

The feature extractor is trained adversarially to move the probability of unknown class away from 0.5 on target examples by utilizing the gradient reversal layer.

*Separate-To-Adapt (STA)*  Liu et al. [46] trained a network that learns jointly from source and target by learning to separate negative (novel) examples from target. The training is divided into two parts. The first part consists of training a multi-binary $G_c|_{c=1}^{|\mathcal{Y}_s|}$ classifier on labeled source data for each class and a binary classifier $G_b$ which generates the weights $w$ for rejecting target samples in the novel class. The second part consists of feature extractor $G_f$, a classifier $G_y$ and domain discriminator $G_d$ to perform adversarial domain adaptation between source and target data in the source label space. $G_y$ and $G_d$ are trained with incorporating weights $w$ predicted by $G_b$ in the first stage.

*Calibrated Multiple Uncertainties (CMU)*  Fu et al. [25] trained a source classifier and a domain discriminator to discriminate the novel class from previously seen classes in target. To train the discriminator network, CMU uses a weighted binary cross entropy loss where $w(x)$ for each example $x$ in target which is the average of uncertainty estimates, e.g. prediction confidence of source classifier. During test time, target data $x$ with $w(x) \geqslant w_0$ (for some pre-defined threshold $w_0$) is classified as an example from previously seen classes and is given a class prediction with source classifier. Otherwise, the target example is classified as belonging to the novel class.

*DANCE*  Saito et al. [59] proposed DANCE which combines a self-supervised clustering loss to cluster neighboring target examples and an entropy separation loss to consider alignment with source. Similar to CMU, during test time, DANCE uses thresholded prediction entropy of the source classifier to classifier a target example as belonging to the novel class.

*Universal Adaptation Networks (UAN)*  You et al. [73] proposed UAN which also trains a source classifier and a domain discriminator to discriminate the novel class from previously seen classes in target. The objective is similar to CMU where instead of using uncertainty estimates from multiple classifiers, UAN uses prediction confidence of domain discriminator classifier. Similar to CMU, at test time, target data $x$ with $w(x) \leqslant w_0$ (for some pre-defined threshold $w_0$) is classified as an example from previously seen classes and is given a class prediction with source classifier. Otherwise, the target example is classified as belonging to the novel class.

For alternative baselines, we experiment with source classifier directly deployed on the target data which may contain novel class and label shift among source classes (referred to as *source-only*). This naive comparison is included to quantify benefits of label shift correction and identifying novel class over a typical $k$-way classifiers.

We also train a domain discriminator classifier for source versus target (referred to as *domain disc.*). This is an adaptation of PU learning baseline[24] which assumes no label shift among source classes. We use simple domain discriminator training to distinguish source versus target. To estimate the fraction of novel examples, we use the EN estimator proposed in Elkan and Noto [24]. For any target input, we make a prediction with the domain discriminator classifier (after re-scaling the sigmoid output with the estimate proportion of novel examples). Any example that is classified as target, we assign it the class $k + 1$. For examples classified as source, we make a prediction for them using the $k$-way source classifier.

Finally, per the reduction presented in Sec. 5, we train $k$ PU classifiers (referred to as *k-PU*). To train each PU learning classifier, we can plugin any method discussed in Sec. A. In the main paper, we included results obtained with plugin state-of-the-art PU learning algorithms. In App. F.8, we present ablations with other PU learning methods.

## F.2  Dataset and OSLS Setup Details

We conduct experiments with seven benchmark classification datasets across vision, natural language, biology and medicine. Our datasets span language, image and table modalities. For each dataset, we simulate an OSLS problem. We experiment with different fraction of novel class prevalence, source label distribution, and target label distribution. We randomly choose classes that constitute the novel target class. After randomly choosing source and novel classes, we first split the training data from each source class randomly into two partitions. This creates a random label distribution for shared classes among source and target. We then club novel classes to assign them a new class (i.e. $k + 1$). Finally, we throw away labels for the target data to obtain an unsupervised DA problem. We repeat the same process on iid hold out data to obtain validation data with no target labels. For main

experiments in the paper, we next describe important details for the OSLS setup simulated. All the other details can be found in the code repository.

For vision, we use CIFAR10, CIFAR100 [40] and Entity30 [61]. For language, we experiment with Newsgroups-20 dataset. Additionally, inspired by applications of OSLS in biology and medicine, we experiment with Tabula Muris [17] (Gene Ontology prediction), Dermnet (skin disease prediction), and BreakHis [66] (tumor cell classification).

**CIFAR10** For CIFAR10, we randomly select 9 classes as the source classes and a novel class formed by the remaining class. After randomly sampling the label marginal for source and target randomly, we get the prevalence for novel class as 0.2152.

**CIFAR100** For CIFAR100, we randomly select 85 classes as the source classes and a novel class formed by aggregating the data from 15 remaining classes. After randomly sampling the label marginal for source and target randomly, we get the prevalence for novel class as 0.2976.

**Entity30** Entity30 is a subset of ImageNet [54] with 30 super classes. For Entity30, we randomly select 24 classes as the source classes and a novel class formed by aggregating the data from 6 remaining classes. After randomly sampling the label marginal for source and target randomly, we get the prevalence for novel class as 0.3942.

**Newgroups-20** For Newsgroups20[2], we randomly select 16 classes as the source classes and a novel class formed by aggregating the data from 4 remaining classes. After randomly sampling the label marginal for source and target randomly, we get the prevalence for novel class as 0.3733. This dataset is motivated by scenarios where novel news categories can appear over time but the distribution of articles given a news category might stay relatively unchanged.

**BreakHis** BreakHis[3] contains 8 categories of cell types, 4 types of benign breast tumor and 4 types malignant tumors (breast cancer). Here, we simulate OSLS problem specifically where 6 cell types are observed in the source (3 from each) and a novel class appears in the target with 1 cell type from each category. After randomly sampling the label marginal for source and target randomly, we get the prevalence for novel class as 0.2708.

**Dermnet** Dermnet data contains images of 23 types of skin diseases taken from Dermnet NZ[4]. We simulate OSLS problem specifically where 18 diseases are observed in the source and a novel class appears in the target with the rest of the 5 diseases. After randomly sampling the label marginal for source and target randomly, we get the prevalence for novel class as 0.3133.

**Tabula Muris** Tabula Muris dataset [17] comprises of different cell types collected across 23 organs of the mouse model organism. We use the data pre-processing scripts provided in [12][5]. We just use the training set comprising of 57 classes for our experiments. We simulate OSLS problem specifically where 28 cell types are observed in the source and a novel class appears in the target with the rest of the 29 cell types. After randomly sampling the label marginal for source and target randomly, we get the prevalence for novel class as 0.6366.

## F.3 Details on the Experimental Setup

We use Resnet18 [33] for CIFAR10, CIFAR100, and Entity30. For all three datasets, in our main experiments, we train Resnet-18 from scratch. We use SGD training with momentum of 0.9 for 200 epochs. We start with learning rate 0.1 and decay it by multiplying it with 0.1 every 70 epochs. We use a weight decay of $5 \times 10^{-4}$. For CIFAR100 and CIFAR10, we use batch size of 200. For Entity30, we use a batch size of 32. In App. F.7, we experiment with contrastive pre-training instead of random initialization.

For newsgroups, we use a convolutional architecture[6]. We use glove embeddings to initialize the embedding layer. We use Adam optimizer with a learning rate of 0.0001 and no weight decay. We use a batch size of 200. We train with constant learning rate for 120 epochs.

---

[2]http://qwone.com/~jason/20Newsgroups/

[3]https://web.inf.ufpr.br/vri/databases/breast-cancer-histopathological-database-breakhis/

[4]http://www.dermnet.com/dermatology-pictures-skin-disease-pictures

[5]https://github.com/snap-stanford/comet

[6]https://github.com/mireshghallah/20Newsgroups-Pytorch

For Tabular Muris, we use the fully connected MLP used in Cao et al. [12]. We use the hyperparameters used in Cao et al. [12]. We use Adam optimizer with a learning rate of 0.0001 and no weight decay. We train with constant learning rate for 40 epochs. We use a batch size of 200.

For Dermnet and BreakHis, we use Resnet-50 pre-trained on Imagenet. We use an initial learning rate of 0.0001 and decay it by 0.96 every epoch. We use SGD training with momentum of 0.9 and weight decay of $5 \times 10^{-4}$. We use a batch size of 32. These are the default hyperparameters used in Alom et al. [2] and Liao [44].

For all methods, we use the same backbone for discriminator and source classifier. Additionally, for PULSE and domain disc., we use the exact same set of hyperparameters to train the domain discriminator and source classifier. For kPU, we use a separate final layer for each class with the same backbone. We use the same hyperparameters described above for all three methods. For OSDA methods, we use default method specific hyperparameters introduced in their works. Since we do not have access to labels from the target data, we do not perform hyperparameter tuning but instead use the standard hyperparameters used for training on labeled source data. In future, we may hope to leverage heuristics proposed for accuracy estimation without access to labeled target data [30].

We train models till the performance on validation source data (labeled) ceases to increase. Unlike OSDA methods, note that we do not use early stopping based on performance on held-out labeled target data. To evaluate classification performance, we report target accuracy on all classes, seen classes and the novel class. For target marginal, we separately report estimation error for previously seen classes and for the novel class. For the novel class, we report absolute difference between true and estimated marginal. For seen classes, we report average absolute estimation error. We open-source our code at `https://github.com/Neurips2022Anon`. By simply changing a single config file, new OSLS setups can be generated and experimented with.

Note that for our main experiments, for vision datasets (i.e., CIFAR10, CIFAR100, and Entity30) and for language dataset, we do not initialize with a (supervised) pre-trained model to avoid overlap of novel classes with the classes in the dataset used for pre-training. For example, labeled Imagenet-1k is typically used for pre-training. However, Imagenet classes overlaps with all three vision datasets employed and hence, we avoid pre-trained initialization. In App. F.7, we experiment with contrastive pre-training on Entity30 and CIFAR100. In contrast, for medical datasets, we leverage Imagenet pre-trained models as there is no overlap between classes in BreakHis and Dermnet with Imagenet.

### F.4 Detailed results from main paper

For completeness, we next include results for all datasets. In particular, for each dataset we tabulate (i) overall accuracy on target; (ii) accuracy on seen classes in target; (iii) accuracy on the novel class; (iv) sum of absolute error in estimating target marginal among previously seen classes, i.e., $\sum_{y \in \mathcal{Y}_s} |\widehat{p}_t(y) - p_t(y)|$; and (v) absolute error for novel fraction estimation, i.e., $|\widehat{p}_t(y = k+1| - p_t(y = k+1)$. Table 5 presents results on all the datasets. Fig. 4 and Fig. 5 presents epoch-wise results.

### F.5 Investigation into OSDA approaches

We observe that with default hyperparameters, popular OSDA methods significantly under perform as compared to PULSE. We hypothesize that the primary reasons underlying the poor performance of OSDA methods are (i) the heuristics employed to detect novel classes; and (ii) loss functions incorporated to improve alignment between examples from common classes in source and target. To detect novel classes, a standard heuristic employed popular OSDA methods involves thresholding uncertainty estimates (e.g., prediction entropy, softmax confidence [73, 25, 59]) at a predefined threshold $\kappa$. However, a fixed $\kappa$, may not for different datasets and different fractions of the novel class. Here, we ablate by (i) removing loss function terms incorporated with an aim to improve source target alignment; and (ii) vary threshold $\kappa$ and show improvements in performance of these methods.

For our investigations, we experiment with CIFAR10, with UAN and DANCE methods. For DANCE, we remove the entropy separation loss employed to encourage align target examples with source examples. For UAN, we remove the adversarial domain discriminator training employed to align target examples with source examples. For both the methods, we observe that by removing the corresponding loss function terms we obtain a marginal improvement. For DANCE on CIFAR10, the

performance goes up from 70.4 to 72.5 (with the same hyperparameters as the default run). FOR UAN, we observe similar minor improvements, where the performance goes up from 15.4 to 19.6.

Next, we vary the threshold used for detecting the novel examples. By optimally tuning the threshold for CIFAR10 with UAN, we obtain a substantial increase. In particular, the overall target accuracy increases from 19.6 to 33.1. With DANCE on CIFAR10, optimal threshold achieves 75.6 as compared to the default accuracy 70.4. In contrast, our two-stage method PULSE avoids the need to guess $\kappa$, by first estimating the fraction of novel class which then guides the classification of novel class versus previously seen classes.

### F.6 Ablation with novel class fraction

In this section, we ablate on novel class proportion on CIFAR10, CIFAR100 and Newsgroups20. For each dataset we experiment with three settings, each obtained by varying the number of classes from the original data that constitutes the novel classes. We tabulate our results in Table 4.

### F.7 Contrastive pre-training on unlabeled data

Here, we experiment with contrastive pre-training to pre-train the backbone networks used for feature extraction. In particular, we initialize the backbone architectures with SimCLR pre-trained weights. We experiment with CIFAR100 and Entity30 datasets. Instead of pre-training on mixture of source and target unlabeled data, we leverage the publicly available pre-trained weights[7]. Table 2 summarizes our results. We observe that pre-training improves over random initialization for all the methods with PULSE continuing to outperform other approaches.

Table 2: Comparison with different OSLS approaches with pre-trained feature extractor. We use SimCLR pre-training to initialize the feature extractor for all the methods. All methods improve over random initialization (in Table 1). Note that PULSE continues to outperform other approaches.

| Method | CIFAR100 | | Entity30 | |
|---|---|---|---|---|
| | Acc (All) | MPE (Novel) | Acc (All) | MPE (Novel) |
| BODA [58] | 37.1 | 0.34 | 52.1 | 0.376 |
| Domain Disc. | 49.4 | 0.041 | 57.4 | 0.024 |
| kPU | 37.5 | 0.297 | 70.1 | 0.32 |
| PULSE (Ours) | 67.3 | 0.052 | 72.4 | 0.002 |

### F.8 Ablation with different PU learning methods

In this section, we experiment with alternative PU learning approaches for PULSE and kPU. In particular, we experiment with the next best alternatives, i.e., nnPU instead of CVIR for classification and DEDPUL instead of BBE for target marginal estimation. We refer to these as kPU (alternative) and PULSE (alternative) in Table 3. We present results on three datasets: CIFAR10, CIFAR100 and Newsgroups20 in the same setting as described in Sec. F.2. We make two key observations: (i) PULSE continues to dominate kPU with alternative choices; (ii) CVIR and BBE significantly outperform alternative choices.

### F.9 Age Prediction Task

We consider an experiment on UTK Face dataset[8]. We create an 8-way class classification problem where we split the age in the following 8 groups: 0–10, 11–20, $\cdots$, 60–70 and > 70. We consider the first 7 age groups in source and introduce age group > 70 into the target data. OSLS continues to

---

[7]For CIFAR100: `https://drive.google.com/file/d/1huW-ChBVvKcx7t8HyDaWTQB5Li1Fht9x/view` and for Entity30, we use Imagenet pre-trained weights from here: `https://github.com/AndrewAtanov/simclr-pytorch`.

[8]`https://susanqq.github.io/UTKFace/`

Table 3: Comparison with different PU learning approaches. 'Alternative' denotes results with employing nnPU for classification and DEDPUL for target marginal estimation instead of 'default' which uses CVIR and BBE.

| Method | CIFAR10 | | CIFAR100 | | Newsgroups20 | |
|---|---|---|---|---|---|---|
| | Acc (All) | MPE (Novel) | Acc (All) | MPE (Novel) | Acc (All) | MPE (Novel) |
| $k$-PU (alternative) | 53.4 | 0.215 | 12.1 | 0.298 | 14.1 | 0.373 |
| $k$-PU (default) | 83.6 | 0.036 | 36.3 | 0.298 | 52.1 | 0.307 |
| PULSE (alternative) | 80.5 | 0.05 | 30.1 | 0.231 | 39.8 | 0.223 |
| PULSE (default) | 86.1 | 0.008 | 63.4 | 0.078 | 62.2 | 0.061 |

Table 4: Comparison with different OSLS approaches for different novel class prevalence. We observe that for on CIFAR100 and Newsgroups20, PULSE maintains superior performance as compared to other approaches. On CIFAR10, as the proportion of novel class increases, the performance of of kPU improves slightly over PULSE for target accuracy.

| Method | CIFAR10 ($p_t(k+1) = 0.215$) | | CIFAR10 ($p_t(k+1) = 0.406$) | | CIFAR10 ($p_t(k+1) = 0.583$) | |
|---|---|---|---|---|---|---|
| | Acc (All) | MPE (Novel) | Acc (All) | MPE (Novel) | Acc (All) | MPE (Novel) |
| BODA [58] | 63.1 | 0.162 | 65.5 | 0.166 | 48.6 | 0.265 |
| Domain Disc. | 47.4 | 0.331 | 57.5 | 0.232 | 68.7 | 0.144 |
| kPU | 83.6 | 0.036 | 87.8 | 0.010 | 89.9 | 0.036 |
| PULSE (Ours) | 86.1 | 0.008 | 87.4 | 0.009 | 83.7 | 0.006 |

| Method | CIFAR100 ($p_t(k+1) = 0.2976$) | | CIFAR100 ($p_t(k+1) = 0.4477$) | | CIFAR100 ($p_t(k+1) = 0.5676$) | |
|---|---|---|---|---|---|---|
| | Acc (All) | MPE (Novel) | Acc (All) | MPE (Novel) | Acc (All) | MPE (Novel) |
| BODA [58] | 36.1 | 0.41 | 41.6 | 0.075 | 50.2 | 0.03 |
| Domain Disc. | 45.8 | 0.046 | 52.3 | 0.092 | 58.7 | 0.187 |
| kPU | 36.3 | 0.298 | 52.2 | 0.448 | 63.9 | 0.568 |
| PULSE (Ours) | 63.4 | 0.078 | 66.6 | 0.052 | 68.2 | 0.088 |

| Method | Newsgroups20 ($p_t(k+1) = 0.3733$) | | Newsgroups20 ($p_t(k+1) = 0.6452$) | | Newsgroups20 ($p_t(k+1) = 0.7688$) | |
|---|---|---|---|---|---|---|
| | Acc (All) | MPE (Novel) | Acc (All) | MPE (Novel) | Acc (All) | MPE (Novel) |
| BODA [58] | 43.4 | 0.16 | 25.5 | 0.645 | 17.7 | 0.769 |
| Domain Disc. | 50.9 | 0.176 | 44.8 | 0.085 | 47.8 | 0.064 |
| kPU | 52.1 | 0.373 | 50.2 | 0.645 | 35.5 | 0.769 |
| PULSE (Ours) | 62.2 | 0.061 | 71.7 | 0.044 | 75.73 | 0.179 |

outperform the $k$PU baseline for novel prevalence estimation. Additionally, for target classification performance of OSLS is similar to k$PU$ baseline (ref. Table 6).

Table 5: *Comparison of PULSE with other methods.* Across all datasets, PULSE outperforms alternatives for both target classification and novel class prevalence estimation. Acc (All) is target accuracy, Acc (Seen) is target accuracy on examples from previously seen classes, and Acc (Novel) is recall for novel examples. MPE (Seen) is sum of absolute error for estimating target marginal among previously seen classes and MPE (Novel) is absolute error for novel prevalence estimation. Results reported by averaging across 3 seeds.

| | CIFAR-10 | | | | | CIFAR-100 | | | | |
|---|---|---|---|---|---|---|---|---|---|---|
| Method | Acc (All) | Acc (Seen) | Acc (Novel) | MPE (Seen) | MPE (Novel) | Acc (All) | Acc (Seen) | Acc (Novel) | MPE (Seen) | MPE (Novel) |
| Source-Only | 67.1 | 87.0 | - | - | - | 46.6 | 66.4 | - | - | - |
| UAN [73] | 15.4 | 19.7 | 25.2 | 1.44 | 0.214 | 18.1 | 40.6 | 14.8 | 1.48 | 0.133 |
| BODA [58] | 63.1 | 66.2 | 42.0 | 0.541 | 0.162 | 36.1 | 17.7 | 81.6 | 0.564 | 0.41 |
| DANCE [59] | 70.4 | 85.5 | 14.5 | 0.784 | 0.174 | 47.3 | 66.4 | 1.2 | 0.702 | 0.28 |
| STA [46] | 57.9 | 69.6 | 14.9 | 0.409 | 0.124 | 42.6 | 48.5 | 34.8 | 0.798 | 0.14 |
| CMU [25] | 62.1 | 77.9 | 41.2 | 0.443 | 0.183 | 35.4 | 46.0 | 15.5 | 0.695 | 0.161 |
| Domain Disc. | 47.4 | 87.0 | 30.6 | - | 0.331 | 45.8 | 66.5 | 39.1 | - | **0.046** |
| $k$-PU | 83.6 | 79.4 | **98.9** | **0.062** | 0.036 | 36.3 | 22.6 | **99.1** | 6.31 | 0.298 |
| PULSE (Ours) | **86.1** | **91.8** | 88.4 | 0.091 | **0.008** | **63.4** | **67.2** | 63.5 | **0.365** | 0.078 |

| | Entity30 | | | | | Newsgroup20 | | | | |
|---|---|---|---|---|---|---|---|---|---|---|
| Method | Acc (All) | Acc (Seen) | Acc (Novel) | MPE (Seen) | MPE (Novel) | Acc (All) | Acc (Seen) | Acc (Novel) | MPE (Seen) | MPE (Novel) |
| Source-Only | 32.0 | 53.5 | - | - | - | 39.3 | 64.4 | - | - | - |
| BODA [58] | 42.22 | 25.9 | 67.2 | 0.367 | 0.189 | 43.4 | 38.0 | 34.1 | 0.550 | 0.167 |
| Domain Disc. | 43.2 | 53.5 | 68.0 | - | 0.135 | 50.9 | 64.4 | **93.2** | - | 0.176 |
| $k$-PU | 50.7 | 22.3 | **94.4** | 0.99 | 0.394 | 52.1 | 57.8 | 42.7 | 0.776 | 0.373 |
| PULSE (Ours) | **58.0** | **54.3** | 72.2 | **0.215** | **0.054** | **62.2** | **65.0** | 83.6 | **0.232** | **0.061** |

| | Tabula Muris | | | | | BreakHis | | | | |
|---|---|---|---|---|---|---|---|---|---|---|
| Method | Acc (All) | Acc (Seen) | Acc (Novel) | MPE (Seen) | MPE (Novel) | Acc (All) | Acc (Seen) | Acc (Novel) | MPE (Seen) | MPE (Novel) |
| Source-Only | 33.8 | 93.3 | - | - | - | 70.0 | 95.8 | - | - | - |
| BODA [58] | 76.5 | 59.8 | 87.0 | 0.200 | 0.079 | 71.5 | 81.8 | 44.0 | 0.163 | 0.077 |
| Domain Disc. | 73.0 | 93.3 | **94.7** | - | 0.071 | 56.5 | 95.8 | **90.4** | - | 0.09 |
| $k$-PU | 85.9 | 91.6 | 83.3 | **0.279** | 0.307 | 75.6 | 71.7 | 86.1 | 0.094 | 0.058 |
| PULSE (Ours) | **87.8** | **94.6** | 88.8 | 0.388 | **0.058** | **79.1** | **96.1** | 76.3 | **0.090** | **0.054** |

| | Dermnet | | | | |
|---|---|---|---|---|---|
| Method | Acc (All) | Acc (Seen) | Acc (Novel) | MPE (Seen) | MPE (Novel) |
| Source-Only | 41.4 | 53.6 | - | - | - |
| BODA [58] | 43.8 | 31.4 | 58.4 | **0.401** | 0.207 |
| Domain Disc. | 40.6 | 53.6 | 82.7 | - | 0.083 |
| $k$-PU | 46.0 | 26.0 | **89.9** | 1.44 | 0.313 |
| PULSE (Ours) | **48.9** | **53.7** | 57.7 | 0.41 | **0.043** |

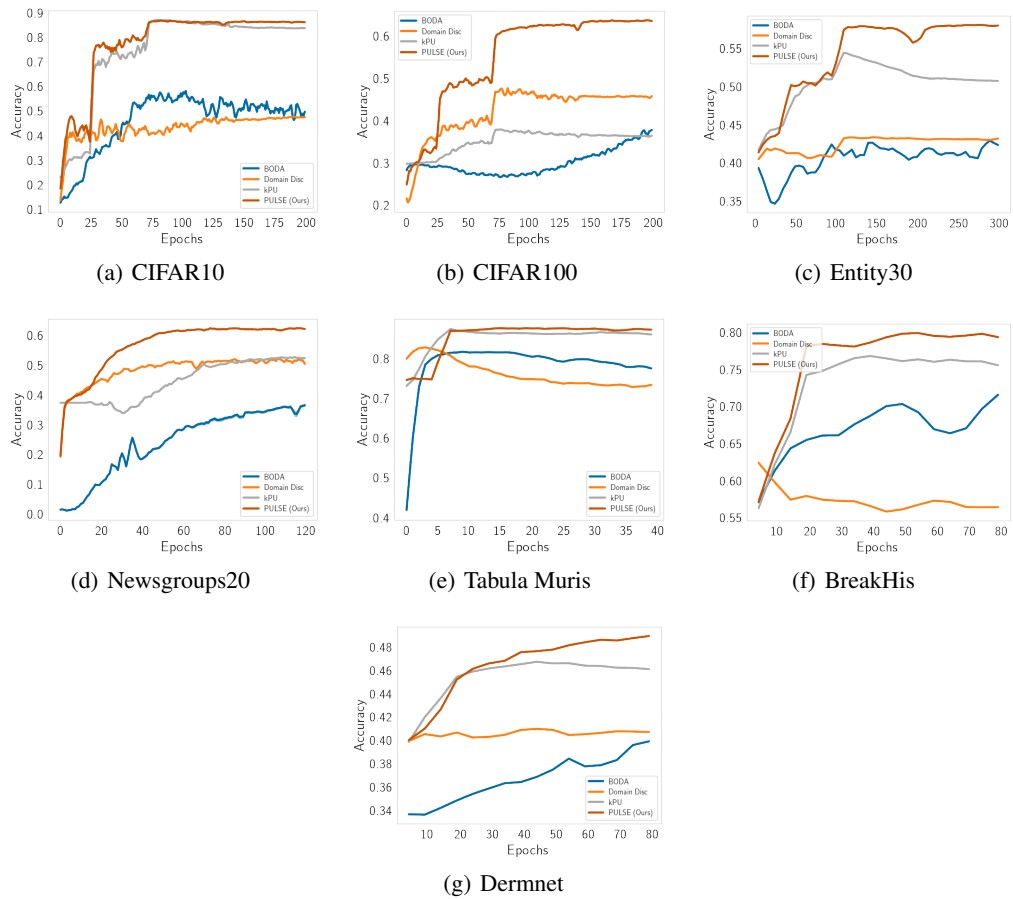

Figure 4: **Epoch wise results for target accuracy.** Results aggregated over 3 seeds. PULSE maintains stable and superior performance when compared to alternative methods.

Table 6: Results on age prediction dataset. We observe that the prevalence of the novel class as estimated with our PULSE framework is significantly closer to the true estimate. Additionally target classification performance of OSLS is similar to that of $k$PU both of which significantly improve over domain discriminator and source only baselines.

|  | **UTK Face** | |
| Method | Acc (All) | MPE (Novel) |
| --- | --- | --- |
| Source Only | 50.1 | 0.11 |
| Domain Disc. | 52.4 | 0.08 |
| kPU | 56.7 | 0.11 |
| PULSE (Ours) | 56.8 | 0.01 |

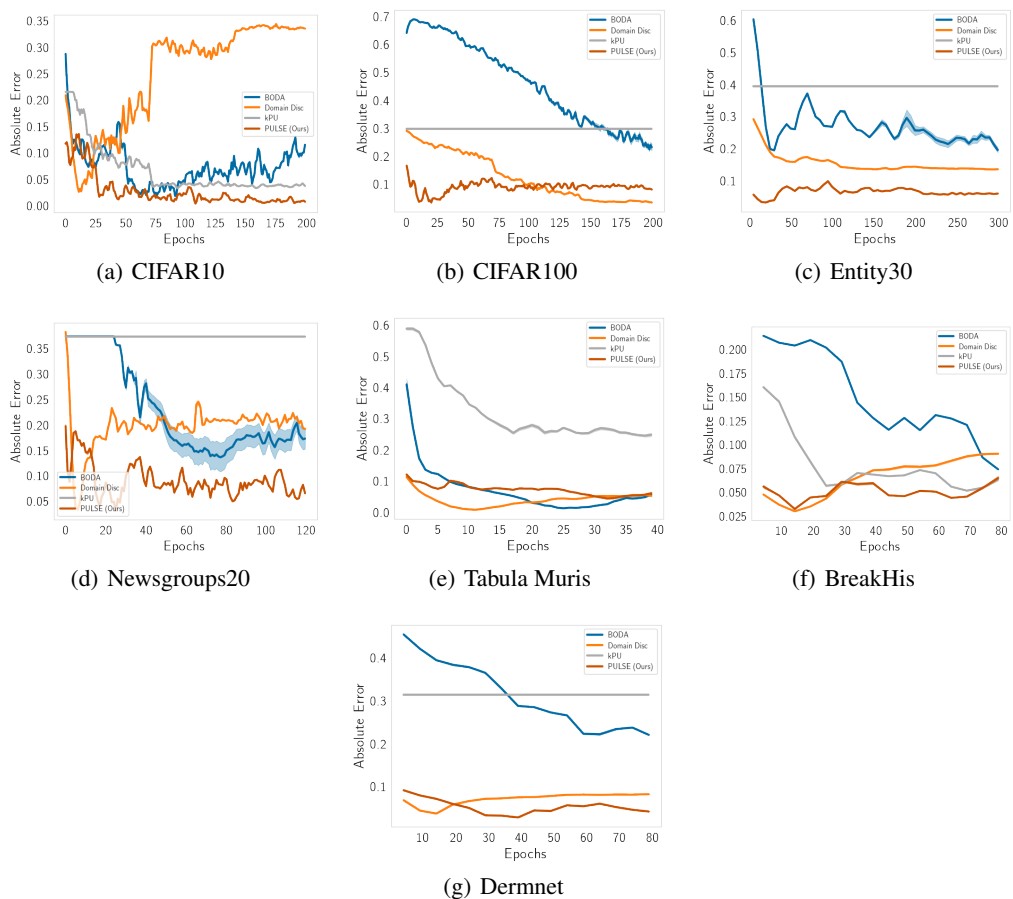

Figure 5: **Epoch wise results for novel prevalence estimation.** Results aggregated over 3 seeds. PULSE maintains stable and superior performance when compared to alternative methods.