# OpenReview forum: "Domain Adaptation under Open Set Label Shift"
_NeurIPS.cc/2022/Conference — NeurIPS 2022 Accept_

### Official Review · Reviewer_dUdt · 2022-07-08

**Rating:** 5
**Confidence:** 3
**Soundness:** 2 fair
**Presentation:** 2 fair
**Contribution:** 2 fair

**Summary:**

This paper introduces domain adaptation under Open Set Label Shift (OSLS). Specifically, it assumes the target has one more novel class that was not previously seen in the source domain while allowing label shifts between source and target domains. This work provides theoretical findings of OSLS, specifically, the necessary condition "weak positivity" and two sufficient conditions "strong positivity" and "separability". The author further proposes a framework to solve OSLS, named PULSE which combines techniques from both areas of (1) positive and unlabeled learning and (2) DA under label shift. The effectiveness of the proposed methods is demonstrated in language, image, and medical datasets.

**Questions:**

1. Does OSLS apply to medical applications such as COVID diagnosis?

The author motivates the OSLS problem by giving a motivating example in the medical domain: the disease proportion changes seasonally and sometime may come with a new disease, like COVID-19. I would like the author to analyze whether the OSLS is suitable for the disease diagnosis problem. For example, let us consider a DA problem where the source domain has two classes, normal people and flu patients while the target domain will have one more class, COIVD patients. How are your identifiability conditions satisfied in this application? Does the PLUSE framework still apply?

2. Does all the 5 tasks in the experiments satisfy "strong positivity"? Could the author include benchmarks not satisfying this condition?

The author claims that "we observe that techniques proposed under strong positivity were empirically stable and outperform other baselines developed under separability. This is intuitive for many benchmark datasets where for each class there exists a subdomain that only belongs to that class." First I want to ask, do all 5 benchmarks used in your experiments all satisfy that there exists a subdomain that only belongs to that class? Second, I would like to point out that samples belonging to different classes are disjoint is the property of object recognition tasks however, there are many other types of tasks that do not satisfy such disjoint property. For example, in age prediction (classify different age groups). It is unlikely that there exist samples in the age group [20,30)'s portrait will never appear in the age group [30,40) or [10,20) since the boundary between different classes is not perfectly clear here. Thus I think it is necessary to include other benchmarks that do not satisfy "strong possibility" for the sake of a more complete evaluation of the proposed method.


**Limitations:**

This work does not raise potential negative societal impact.

**Strengths And Weaknesses:**

originality
---
The paper focuses on a special case of the open set domain adaptation where the target domain has one novel class that is not previously unseen in the source domain. The big concept of open set DA is not novel while the special case is well studied yet, thus can be considered novel. The paper delivers an identifiability analysis of the OSLS which is novel.

quality
---
The paper provides sold theoretical analysis as well as extensive experiments on 5 datasets across multiple application domains.


Minor issue, typos:

1. Line 46: … the matrix submatrix…
2. Line 67: double periods after "(Sec.7)"
3. Line 154: \frac{1}{p_t(y=k+1)} is missing in the closed form of p_t(x|y=k+1)
4. The paper seems not define what the metric "novel prevalence estimation" is.

clarity
---
The paper has many contents and is pretty dense. The majority of the paper is well-written and clear. However, I find the method section a bit obfuscated. One reason is that there is no headings or subsection title to remind the reader what is the focus of each paragraph. Another reason is that the author seems to jump to the details too quickly. The logic of the current text is kind of linear. I would like to suggest the author organize section 6 and each of its subsections to have an overview-and-details structure.

significance
---
The paper does a good job on the specific problem it aims to solve. The analysis is solid and the empirical results are illustrative. My main concern is that the paper's significance is constrained by the practicality of the proposed problem. The problem setting is a bit artificial to me. I am not sure how practical it is that we assume the target domain has exactly one novel class unseen. The experiment is kind of synthetic/semi-synthetic since the author chooses source and novel classes randomly.

---

> ### Author Response · Authors · 2022-08-02
> **Response to R4**
>
> Thank you for your positive assessment and constructive feedback on our work.
>
> **However, I find the method section a bit obfuscated. One reason is that there is no headings or subsection title to remind the reader what is the focus of each paragraph... I would like to suggest the author organize section 6 and each of its subsections to have an overview-and-details structure.**
> Thanks for your suggestion. As per your suggestions, we have re-structured and re-written Section 6 of the paper to improve the exposition. We added a high-level description of the algorithm with separate paragraph titles detailing the algorithm.
>
> **My main concern is that the paper's significance is constrained by the practicality of the proposed problem. The problem setting is a bit artificial to me. I am not sure how practical it is that we assume the target domain has exactly one novel class unseen. The experiment is kind of synthetic/semi-synthetic since the author chooses source and novel classes randomly.**
>
> We would like to make several clarifications. First, in our work, the novel class can be a union of multiple classes. The goal of our work is to reject all of them into a separate class (i.e. k+1) which can further trigger appropriate auditing responses. For example, if the prevalence of the novel class in the unlabeled target data is significant, the practitioner can choose to acquire labels for the identified examples from the novel class.
>
> Second, despite its simplicity, the OSLS setting is of significant practical relevance. For example, biologists often have a labeled database with known cell types (source) and an unlabeled database of tissue cells (target) which may contain novel cell types [Cao et al 2019, Elkan and Noto 2008]. They can use the data to train a classifier model and then deploy the model to not only identify cell types in a target tissue but also to discover previously unknown cell types. The OSLS problem can also routinely appear in medical diagnosis where disease cause symptom [Lipton et al. 2019]. Along with changing prevalence of known diseases, novel diseases can appear over time.
>
> This has been the motivation behind our choices of  BreakHis (tumor cell classification)  and DermNet (skin disease classification) datasets. For example, DermNet is motivated by scenarios where doctors have labeled data of several skin diseases (source) and they may wish to deploy a model trained on source data on unlabeled data where along with changing prevalence of previously seen diseases images with new diseases or no diseases may appear.
>
> Third, similar to the previous work in simulating open set domain adaptation problems and due to the lack of benchmark datasets with previously unseen classes, we randomly divide the classes in datasets we consider into seen and novel classes. By repeating experiments for different novel fractions and with different seeds, we hope to cover numerous diverse settings.
>
> Overall, instead of working on general domain adaptation problems which are mathematically ill-posed, our goal is to expand the umbrella of structured distribution shift settings which allows for the development of principled machinery. As discussed in our work, the OSLS setting introduced in our work is strictly more general than the structured setting of label shift and PU learning settings widely explored in the past literature [56, 45, 4, 1, 27, 77, 24, 22, 63, 36, 6, 7, 23, 21].
>
> **Does OSLS apply to medical applications such as COVID diagnosis?**
>
> We agree that the OSLS assumption is strong and may not strictly hold exactly in practice. However, as discussed before, we believe that rigorously understanding these idealized settings is a fundamental building block toward more complex settings. Moreover, in many important problems (e.g. medical diagnosis during an epidemic), OSLS is nevertheless a useful model because the prevalence is likely to change faster than conditional probabilities of symptoms given disease.
>
> Strong positivity may be satisfied if we aim to distinguish symptomatic COVID from flu and normal people (e.g. if fever is not common with some flu patients but common with a large population with symptomatic COVID, then it may give us a sub-domain specific to the flu). Depending on the task, strong positivity can be best understood by the practitioner. As discussed above, a more relevant example is distinguishing cell types in biology where an unlabeled database of tissue cells (target) may contain novel cell types, and a labeled database with known cell types (source) can be used to formulate it into OSLS problem.

---

> > ### Author Response · Authors · 2022-08-02
> > **continued response**
> >
> > **Does all the 5 tasks in the experiments satisfy "strong positivity"? Could the author include benchmarks not satisfying this condition?**
> > Thanks for this suggestion. We have now included experiments on an age-prediction task (UTK-Face) in Appendix F.9 in Table 6. We observe that the prevalence of the novel class as estimated with our PULSE framework is significantly closer to the true estimate. Additionally target classification performance of OSLS is similar to that of $k$PU both of which significantly improve over domain discriminator and source only baselines.
> >
> > | Method  |   Acc | MPE|
> > |--------------|-------|------|
> > | Source Only  |   50.1| 0.11 |
> > | Domain Disc. |   52.4 | 0.08 |
> > | kPU          |   56.7  | 0.11 |
> > | PULSE (Ours) |  56.8  | 0.01 |
> >
> >
> >
> >
> > **Minor issue, typos**
> > Thanks for catching the typos. We have fixed them all in the updated version.
> >
> > **The paper seems not to define what the metric novel prevalence estimation is.**
> > For novel class prevalence estimation, we report the absolute difference between the true and estimated prevalence of the novel class in the target (Lines 306-307).
> > For previously seen classes, we report the $\ell_1$ difference between the estimated marginal and true label marginal among previously seen classes in the target.

---

> ### Author Response · Authors · 2022-08-08
> **Checking in**
>
> Hi, since we haven't heard back and the window for discussion is coming to a close we just wanted to check in to see if our replies and improvements to the paper have successfully addressed your concerns or if there is anything else that we contribute in the remaining time to improve the draft to your satisfaction.

---

### Official Review · Reviewer_d8bB · 2022-07-08

**Rating:** 6
**Confidence:** 4
**Soundness:** 2 fair
**Presentation:** 2 fair
**Contribution:** 2 fair

**Summary:**

The goal of this paper is to solve the open set domain adaptation under the label shift setting. The authors proposed a PU learning-based framework to first estimate the label shift and then classify the novel class. Moreover, the authors also gave sufficient and necessary conditions to the open set label shift in order to make the target label marginal identify. The experimental results showed that the PULSE framework could achieve great performance improvement.

**Questions:**

1) The authors need to better organize the paper and improve the mathematical notations, e.g. in line 48 "the matrix submatrix" is not so clear.


2) For the extended BBE algorithm 2, which is the importance weight w? ($\hat{c}_j$ I assume), the author should be more clear.

3) Since the key point in this paper is to estimate the target label marginal distribution using source data, this process relies heavily on Theorem 1 to guarantee the target domain estimation on the source domain. As we can find from Theorem 1, the threshold would be very important, so the author should discuss more about the threshold e.g. is that a hyperparameter? Or is there any trade-off between varying the threshold with the performance on target data? I also wonder that is that directly using the source classifier for target in Algorithm 2 line 6 is guaranteed by this Theorem?

4) Since the PULSE framework is built on the strong positive condition to guarantee the target inevitability, I found no regularization for the data in this paper, so the author also needs to explain why their algorithm can achieve this strong positive condition.

5) In algorithm 2 , the author should expand the loss function in line 6 and line 10 to make it more clear. And The author should explain more on the connection between the $\alpha$ with the novel target and why they are equal and can be derived this way (why don't use the estimation from algorithm 2 but use algorithm 4 instead in algorithm 3, however, I found the resample based estimation should be derived from algorithm 2 as described in line 805).

6) Both in algorithm 2 and algorithm 4, the z is ambitious in the paper, are they discrete or continuous variables, and do we need to compute all of the values from 0 to 1 for estimation? Will that increase the computation burden? Also, for the c in algorithm 2 line 6, do they have the connections with the threshold in Theorem 1 since their notation is all c?

7) How about the results from the general domain adaptation datasets like USPS, OFFICEHOME, DOMAINNET etc. The experiment setup is not convincing, the random choice for source and target in small datasets (like cifar) may make the source and target only have the limited domain discrepancy, which in return limit the domain adaptation problems back into the novel class detection problems under PU setup. Since the domain discrepancy, the author proposed in the paper is mostly between the positive samples and the unlabeled samples, and unlabeled samples are consists of the combination of both positive and negative samples as described in CVIR, I have little concern on if this problem would degrade into open-set classification (since the domain discrepancy is limited) instead of the open-set domain adaptation as we can find in the experiments.

8) In the paper line 188 section 5, why only satisfying the strong positive condition the OSLS can be reduced into K PU? It seems most of the OSLS problem can all be reduced into K PU problems that way in equation 2.

**Limitations:**

1) This paper needs to be well organized.
2) Some details need to be completed as described in the questions.
3) The experiments in this paper is not convincing.


**Strengths And Weaknesses:**

1) The idea of combing PU problems with the OSLS is interesting, the authors used an ingenious way to merge the PU problems into the OSLS. The reduction from the k-PU to a single PU is attractive (equation 3).

2) The definitions and theorems are straightforward and easy to be understood, and the mathematics proof seems solid.

3) The author used a two-stage method to separately estimate the label shift and to identify the novel class, which seems effective in solving both the domain adaptation and novel class identification. The results of the PULSE seems good.

4) I have no doubts in the originality to the best of my knowledge, the quality of this paper is good, some more details need to be discussed for clarity as I stated later, and the contribution of this paper has some significance to the OSLS regime.

---

> ### Author Response · Authors · 2022-08-02
> **Response to R3**
>
> We thank the reviewer for their detailed comments and positive assessment.
>
> **Clarification on typos. The authors need to better organize the paper and improve the mathematical notations, e.g. in line 48 "the matrix submatrix" is not so clear.**
>
> We have fixed the typos. Thanks for catching them.
>
> **For the extended BBE algorithm 2, which is the importance weight w? (c^j I assume), the author should be more clear.**
> Algorithm 2 estimates the label shift among previously seen classes in the source which is denoted with $\widehat p_t^\prime (y)$ in Algorithm 1, step 3, and in the output of Algorithm 2. We have clarified this in the main text.
>
> **As we can find from Theorem 1, the threshold would be very important, so the author should discuss more about the threshold e.g. is that a hyperparameter? Or is there any trade-off between varying the threshold with the performance on target data?**
>
> PULSE builds on top of BBE proposed in Garg et al. 2021b for target marginal estimation. In a PU learning problem (OSLS with k=1), given positive and unlabeled data, BBE estimates the fraction of positives in unlabeled in the push-forward space of the classifier. In particular, instead of operating in the original input space, BBE maps the inputs to one-dimensional outputs (i.e., a score between zero and one) which is the predicted probability of an example being from the positive class. Based on your suggestion, we have now included an intuitive description of the BBE algorithm.
>
> Efficacy of BBE relies on the existence of a threshold on probability scores assigned by the classifier such that the examples mapped to a score greater than the threshold are *mostly* positive. This is referred to as the *top bin property*. If such a threshold exists, BBE algorithm recovers (or learns) such a threshold with $1\sqrt{n}$ guarantees as shown in Theorem 3 and 5. In more general cases when such a threshold doesn’t exist, our guarantees capture the tradeoff due to the proportion of negative examples in the top bin (bias) versus the proportion of positives in the top bin (variance).
>
> To elaborate, BBE minimizes the upper confidence bound in Step 6 of Algorithm 3 (or Step 3 of Algorithm 4) which provides us with the guarantee in Theorem 3 (extension of Theorem 1 from Garg et al. 2021b). Importantly, this theorem shows that minimizing the upper confidence bounds allows us to **learn** the threshold. Note that this threshold is not a hyperparameter and the tightness of recovering this threshold depends on hyperparameters $\gamma$ and $\delta$. Varying the $\delta$ used for high probability bound could provide some trade-off between the tightness of the bound and the associated guarantee. However, as highlighted in Garg et al. 2021b, the trade-off is insignificant and we use the same values as in Garg et al. 2021b.
>
> **I also wonder that is that directly using the source classifier for target in Algorithm 2 line 6 is guaranteed by this Theorem?**
> In general scenarios, our guarantees in Theorem 3 capture the tradeoff due to the proportion of negative examples in the top bin (bias) versus the proportion of positives in the top bin (variance). Additionally (as you suggested), we provide empirical evidence to the top bin property with a source classifier in Figure 2. We elaborate more on this experiment in Appendix D.1.
>
> **Since the PULSE framework is built on the strong positive condition to guarantee the target inevitability, I found no regularization for the data in this paper, so the author also needs to explain why their algorithm can achieve this strong positive condition**
> We think there may be some misunderstanding here. Strong positivity is an assumption on the data and not on the underlying algorithm. Put differently, strong positivity lays down the conditions that the underlying source and target data should satisfy *in order for it to even be possible (for any algorithm)* to learn the parameters of interest in the OSLS problem.
>
> **In algorithm 2 , the author should expand the loss function in line 6 and line 10 to make it more clear.**
> Thanks for your suggestion. We have now made this clear in Appendix C. In Algorithm 3, the loss function can be any standard classification loss. As mentioned in Section 2 (Lines 124-128) we use cross-entropy loss.

---

> > ### Author Response · Authors · 2022-08-02
> > **continued response**
> >
> > **And The author should explain more on the connection between the α with the novel target and why they are equal and can be derived this way (why don't use the estimation from algorithm 2 but use algorithm 4 instead in algorithm 3, however, I found the resample based estimation should be derived from algorithm 2 as described in line 805).**
> >
> > Thanks for your suggestion. We have improved the exposition in Section 6 of the paper to clarify this confusion.
> >
> > We use Algorithm 2 to only estimate the relative proportion of previously seen classes (i.e., all classes except the novel class). As mentioned in Section 5 (and shown in our experiments), using Algorithm 2 to directly estimate the proportion of the novel class as $\widehat p_t (y = k+1 ) = 1 - \sum_{j = 1}^k \widehat p_t( y =j) $ leads to significant under-estimation of the novel proportion due to error accumulation (because of over-estimation bias in mixture proportion in PU learning algorithms).
> >
> > Hence, to estimate the proportion of the novel class we use Algorithm 4. In particular, after estimating the label shift among source classes with Algorithm 2 (i.e., relative proportion $\frac{\widehat p_t(y = j)}{\sum_{i =1}^{k} \widehat p_t( y = i}$ for all $j \in Y_s$), we construct label shift corrected source distribution $p_s^\prime (x)$. We leverage this to obtain a single PU learning problem which we solve using Algorithm 3. This is based on the reduction presented in equation 3. We estimate the proportion of the novel class with Algorithm 4 and the corresponding guarantee is presented in Appendix D.3 (Theorem 5).
> >
> > **Both in algorithm 2 and algorithm 4, the z is ambitious in the paper, are they discrete or continuous variables, and do we need to compute all of the values from 0 to 1 for estimation? Will that increase the computation burden?**
> >
> > We have now clarified the notion Z used in Algorithms 2 and 4 in Appendix C. $Z$ simply denotes the output of the underlying classifier on input $X$. For estimating the proportion of different classes, instead of operating in the input space, we operate in the output space (push-forward) of the classifier. $Z$’s capture the prediction probability and hence are continuous variables that take values between [0,1]. The computation overhead is negligible as the time required to compute Z’s is just the time required for a single forward pass with the underlying classifier.
> >
> > **Also, for the c in algorithm 2 line 6, do they have the connections with the threshold in Theorem 1 since their notation is all c?**
> >
> > Yes, they are related. In fact, in the proof of our Theorem 3 (formal statement of Theorem 1), we show that $\hat c_j$ concentrate to $c^*_j$ (defined in Theorem 3). Intuitively, $c^*_j$ captures the threshold on [0,1] for the top bin such that the ratio of the fractions of positive and unlabeled points receiving scores above the threshold $c_j^*$ is minimized. With Algorithm 2, we aim to ideally estimate these $c_j^*$ for the source classifier to estimate the relative proportion of previously seen classes in the target. With Algorithm 4, we aim to estimate the threshold to estimate the prevalence of the novel class in the target.

---

> > > ### Author Response · Authors · 2022-08-02
> > > **continued response**
> > >
> > > **How about the results from the general domain adaptation datasets like USPS, OFFICEHOME, DOMAINNET etc.. Since the domain discrepancy, the author proposed in the paper is mostly between the positive samples and the unlabeled samples ... I have little concern on if this problem would degrade into open-set classification (since the domain discrepancy is limited) instead of the open-set domain adaptation as we can find in the experiments**
> > >
> > > We think there may be some misunderstanding here. Domain adaptation problems arise when the source distribution $p_s(x,y)$ shifts to a target distribution $p_t(x,y)$. In a closed-set setting, the distribution shift can be induced due to (i) covariate shift where $p(x)$ shifts but p(y|x) remains invariant from source to target; or (ii) label shift where $p(y)$ shifts but $p(x|y)$ remains invariant from source to target (refer to Kun Zhang et al. 2013 [78]).
> > >
> > > OSLS setting extends the label shift (the latter) setting to scenarios where along with shifts in label distribution among source classes, previously unseen classes may be observed. Note that even without the novel class, there is a distribution shift among previously seen classes due to shifting prevalences of common classes (i.e. label shift).
> > >
> > > Additionally without catering to the label shift among source classes and ad-hoc simplification to  directly distinguish previously seen versus novel classes doesn’t work in practice. We empirically demonstrate this by showing that simple domain discriminator-based baselines perform poorly in the OSLS settings.
> > >
> > > To summarize, in our work, we focus on label shift (changing label prevalence) instead of covariate shift (changing appearance) among previously seen classes. Our goal is to expand the umbrella of structured distribution shift settings which allows for the development of principled machinery instead of working on general domain adaptation problems which are mathematically ill-posed. As discussed in our work, the OSLS setting introduced in our work is strictly more general than the structured setting of label shift and PU learning settings widely explored in the past literature [56, 45, 4, 1, 27, 77, 24, 22, 63, 36, 6, 7, 23, 21].
> > >
> > > We hope that our work provides solid ground for principled algorithmic developments to further expand the umbrella of structured shifts. In particular, one may extend the OSLS problem to more general settings for future work where along with shifting label distribution from source to target, p(x|y) can also deviate from source to target within some divergence constraint. As a first step, we performed preliminary experiments on the FMoW dataset from Koh et al. 2021 [3] (see common response).
> > >
> > >
> > >
> > > **In the paper line 188 section 5, why only satisfying the strong positive condition the OSLS can be reduced to K PU? It seems most of the OSLS problems can all be reduced into K PU problems that way in equation 2.**
> > > Yes, mathematically any OSLS problems can be thought of as k-PU problems as per eq (2). However, for the identifiability of each of these PU problems, we need the irreducibility assumption (Davis and Bekkar 2020). Put simply, for individual PU problems defined for source classes $j \in Y_s$, we need the existence of a sub-domain $X_j$ such that we only observe examples for that class j in $X_j$. Collectively $X_j$ gives us the $X_sp$ needed for strong positivity. We have clearly discussed this in Appendix A.1.

---

> ### Author Response · Authors · 2022-08-08
> **Checking in**
>
> Hi, thanks again for your thoughtful review. We would just like to check in to see whether our replies and improvements to the manuscript have successfully addressed your key concerns. Can you please let us know if you are moved to improve your assessment or if there are any other concerns that we might address in the remaining time that could improve the paper to your satisfaction? Thanks!

---

> > ### Comment · Reviewer_d8bB · 2022-08-08
> > **Reply on the rebuttal**
> >
> > Hi,
> > Thanks for the reply, your reply has covered most of my main concerns, and I will raise my assessment.

---

### Official Review · Reviewer_Pd4h · 2022-07-11

**Rating:** 7
**Confidence:** 3
**Soundness:** 4 excellent
**Presentation:** 3 good
**Contribution:** 3 good

**Summary:**

This paper scopes the common Open Set Domain Adaptation problem to specifically consider Open Set Label Shift (OSLS) problems, where $p(x|y)$ is constant, but the class proportions may change between source and target, and there may be a new unseen class added in testing. In this setting, the goal is to identify instances of the unseen class, while also performing adequately on the previously seen classes. The paper proposes a new framework, PULSE, which combines classical Positive and Unlabeled Learning techniques and label reweighting techniques in order to tackle this new problem. The method is empirically shown to perform well in a variety of OSLS problems, and theoretical analysis is conducted to create sufficient conditions for identifiability of the OSLS problem.

**Questions:**

* How robust is PULSE expected to be in general OSDA problems, not just label shift?

**Limitations:**

   It would be worthwhile to have a further discussion on limitations of PULSE with regards to the amount of labeled data of the target domain - for example, with no labels it would be difficult to approximate what the label shift is. It could also be useful to discuss the types of models or datasets that PULSE is expected to work on (e.g. does it perform worse as the number of classes increase, the computational efficiency with regards to larger models, etc.); and that it inherits limitations from its particular stages (e.g. any limitations of importance weighted label shift or CVIR/BBE).

**Strengths And Weaknesses:**

* **Originality**
    To the best of my knowledge, this is the first time I have seen the scoping of the open set domain adaptation problem. By focusing purely on label shift, the authors are able to create some novel theoretical results as well as a highly performant framework for tackling the OSLS problem. In particular, their framework combines two standard techniques to great effect; class reweighting to handle the label shift and PU techniques to handle the open set nature of the problem.
* **Quality**
    The claims are well-substantiated both theoretically and empirically, and the results are impressive. The authors perform a detailed evaluation on existing open source domain adaptation methods as well as a slight ablation study by performing standard PU techniques without any label reweighting.
* **Clarity**
    The paper is mostly well organized and written. A worthwhile addition in the related work section would be to make it more clear exactly who your setting differs from each of the main categories that are defined there. Further, it may be worth a few sentences relating why identifiability is useful and related to the remainder of the paper; Section 4 felt significantly out of place compared to the rest of the paper.
* Some minor nitpicks
     * line 48 - matrix submatrix appears to be a typo.
     * line 257 should likely be a heading
* **Significance**
    By scoping the OSDA problem to focus on label shift, the authors were able to show relatively significant gains compared to standard OSDA methods - this kind of scoping appears quite fruitful for other researchers to build off of. The theoretical contributions in the paper are strong evidence that their framework, PULSE, is able to perform quite well on a variety of scenarios which leaves it as a valuable benchmark for future work.

---

> ### Author Response · Authors · 2022-08-02
> **Response to R2**
>
>
> We thank the reviewer for their positive and thoughtful feedback and for championing our paper.
>
> **A worthwhile addition in the related work section would be to make it more clear exactly who your setting differs from each of the main categories that are defined there**
> Thanks for your suggestion. We have added a few lines highlighting the distinction. Here, we lay out the main differences.  First, DA methods do not handle previously unseen classes in the target. Second, while OSDA methods handle previously unseen classes in target, existing OSDA methods are heuristic in nature, addressing settings where the right answers seem intuitive but are not identified mathematically. Third, PU learning is a base case of OSLS, i.e., when k=1.
>
> **Further, it may be worth a few sentences relating why identifiability is useful and related to the remainder of the paper; Section 4 felt significantly out of place compared to the rest of the paper.**
> Identifiability, in particular sufficient conditions, are needed to understand the nature of assumptions that our datasets (labeled source and unlabeled target) shall satisfy to allow us to tackle the OSLS problem. Additionally, the constructive solution for identifiability under sufficient conditions allows us to start developing practical algorithms to tackle the problem.
>
> **How robust is PULSE expected to be in general OSDA problems, not just label shift?**
> Per standard impossibility results [8], a single domain adaptation method can not handle different kinds of distribution shift problems that may arise. This is empirically corroborated in Sagawa et al. 2022 [57], where no single method provided consistent gains over the ERM baseline.
>
> Thus, instead of working on general domain adaptation problems which are mathematically ill-posed, our goal is to expand the umbrella of structured distribution shift settings which allows for the development of principled machinery. As discussed in our work, the OSLS setting introduced in our work is strictly more general than the structured setting of label shift and PU learning settings widely explored in the past literature [56, 45, 4, 1, 27, 77, 24, 22, 63, 36, 6, 7, 23, 21].
>
> We hope that our work provides solid ground for principled algorithmic developments to further expand the umbrella of structured shifts. In particular, one may extend the OSLS problem to more general settings for future work where along with shifting label distribution from source to target, p(x|y) can also deviate from source to target within some divergence constraint.
>
> As a first step, we performed preliminary experiments on FMoW dataset from Koh et al. 2021 [39] (see common response).

---

> > ### Author Response · Authors · 2022-08-02
> > **continued response**
> >
> > **Limitations of PULSE**
> >
> > Thanks for your suggestion to include the limitations of PULSE. We have added limitations of PULSE in Appendix C.2. We will include them in the main paper in the final version.
> >
> > **It could also be useful to discuss the types of models or datasets that PULSE is expected to work on**
> >
> > Since PULSE inherits the limitations of CVIR and BBE, it is expected to work when the black-box classifiers (i.e., source classifier and domain discriminator classifier) identify an almost pure top bin, i.e., they identify a threshold on the probability score such that examples that get a score higher than the threshold are mostly positive. While this property seems to be satisfied across different datasets spanning different modalities and applications (CIFAR 10 illustrations in Figure 2 in appendix), failure to identify an almost pure top bin can degrade the performance of BBE and hence, our PULSE framework.
> >
> >  **the computational efficiency with regards to larger models**
> >
> > Across our experiments, to adapt the source classifier to the target, we train the domain-discriminator classifier for the same number of epochs as we use to train the source classifier. Hence, as with the typical unsupervised domain adaptation methods, the compute time required is approximately doubled.
> >
> > **the amount of labeled data of the target domain - for example, with no labels it would be difficult to approximate what the label shift is.**
> >
> > We do not need labeled data from the target domain to estimate the label shift. We only need labeled data from the source and unlabeled data from the target to (i) estimate the target marginal (i.e., label shift among source classes and prevalence of the novel class). Our rates in Theorem 1 (and Theorem 5) hint at the number of unlabeled samples from the target required to get good estimates.

---

> > > ### Comment · Reviewer_Pd4h · 2022-08-06
> > > **Thank you**
> > >
> > > Thank you for the detailed response addressing many of my concerns! I am in agreement with these responses, and appreciate the work you have done to integrate these changes into the paper - I believe the paper is much clearer narratively for it.
> > >
> > > I have no other comments or questions for now, but will let you know if I do.

---

### Official Review · Reviewer_iwb5 · 2022-07-11

**Rating:** 6
**Confidence:** 3
**Soundness:** 3 good
**Presentation:** 2 fair
**Contribution:** 3 good

**Summary:**

This paper introduces the problem of domain adaptation under open set label shift (OSLS), where the class-conditional distributions p(x|y) are domain-invariant, and p(y) can change.  Domain Adaptation under label shift and positive-unlabeled (PU) learning can be all considered as special cases of OSLS.   This paper also provides the identifiablity of OSLS, including necessary condition (weak positivity) and sufficient conditions (strong positvity). Under the strong positivity condition, the OSLS problem can be broken into k PU problems.  The PU learning algorithms cannot scale to datasets with large number of classes because of error accumulation. This paper then proposes PULSE framework to solve this problem by exploiting the joint structure of the problem with source-class re-sampling. Experiments across 7 semi-synthetic benchmarks shows that the proposed PULSE consistently outperform OSDA baselines.

**Questions:**

- All experimental results are semi-synthesized and are on relative small datasets, such as CIFAR-10, CIFAR-100. It is good to have results on some popularly used datasets for domain adaptation such as DomainNet.
- Make the writing better



**Limitations:**

- The benchmarks used in this paper are all semi-synthesized, and thus lack proof from real-world application
- The paper mentions that "In future work, we hope to bridge the gap between the necessary and sufficient identifiability conditions."


**Strengths And Weaknesses:**

Strengths
- Open Set Label Shift (OSLS) an interesting problem, assuming p(x|y) is the same and p(y) is changing.
- Theories on identifiablity of the problem and convergence analysis
- New algorithm PULSE to solve the problem
- Good experimental results

Weakness
- All experimental results are semi-synthesized and are on relative small datasets, such as CIFAR-10, CIFAR-100. It is good to have results on some popularly used datasets for domain adaptation such as DomainNet.
- The writing is not easy to follow, the references (e.g., Algorithm 2&3) switch from main paper and supplementary materials without clear instruction. There are also some minor grammar errors including in the formula (line 153)

---

> ### Author Response · Authors · 2022-08-02
> **Response to R1**
>
> We thank the reviewer for their constructive comments and positive assessment. We are glad that you find the OSLS problem setting interesting.
>
> **All experimental results are semi-synthesized and are on relative small datasets, such as CIFAR-10, CIFAR-100. It is good to have results on some popularly used datasets for domain adaptation such as DomainNet.**
> As per your suggestion and the suggestion of Reviewers Pd4h and dUdt, we have included additional experiments on the UTK Face dataset (age prediction from images) and FMOW dataset (please see common response above). Additionally in our experiments, we have large-scale datasets like Entity30 (a subset of Imagenet), BreakHis (tumor cell classification), and DermNet (skin disease classification). All these three datasets are DomainNet scale datasets (in terms of input size and the number of images per class).
>
> Our work is primarily focused on distribution shift settings where (i) p(x|y) remains invariant across common classes in source and target; (ii) novel classes can show up in the target.  Due to the nature of the problem setting in OSLS, we do not include experiments on DomainNet across different domains (e.g. sketch, real images) which doesn’t allow exploration of structured shifts. Instead of working on general domain adaptation problems which are mathematically ill-posed, our goal is to expand the umbrella of structured distribution shift settings which allows for the development of principled machinery. As discussed in our work, the OSLS setting introduced in our work is strictly more general than the structured setting of label shift and PU learning settings widely explored in the past literature [56, 45, 4, 1, 27, 77, 24, 22, 63, 36, 6, 7, 23, 21].
>
> Our work takes the first step in tackling the OSLS problem for large-scale datasets with deep learning. We leave the investigations on extending the OSLS problem to settings where along with shifting label distribution from source to target, p(x|y) can also deviate from source to target with some divergence constraints for future work. As a first step, we performed preliminary experiments on FMoW dataset from Koh et al. 2021 [3] (see common response).
>
> **The writing is not easy to follow, the references (e.g., Algorithm 2&3) switch from the main paper and supplementary materials without clear instruction.**
> Thanks for the feedback. We have significantly updated the manuscript presentation in Section 6 with clear references to Algorithms 2 and 3 in the main paper. Algorithms 2 and 3 are built on top of the BBE and CVIR procedure proposed in previous work. Due to space constraints, we only include the description necessary to understand the PULSE framework by largely treating the BBE and CVIR procedures as black boxes. We have included a detailed description of these algorithms in Section C of the appendix.
>
> **The paper mentions that "In future work, we hope to bridge the gap between the necessary and sufficient identifiability conditions.”**
> Yes, as mentioned in the conclusion and future work, we hope to bridge the gap between the necessary and sufficient identifiability conditions. In our work, we illustrate the existing gap with examples (Examples 1 and 2 in App B.1) in a tabular setting.

---

> ### Author Response · Authors · 2022-08-08
> **Checking in**
>
> Thanks again for your thoughtful review! Since the discussion window is closing, we just wanted to check in to see if our replies successfully addressed your primary concerns or if there is anything else that we can add in the remaining time that might improve the paper such that you might be moved to increase your score. Please let us know if there is anything else that we can do on our end.

---

### Author Response · Authors · 2022-08-02
**Common Response**

We would like to thank the reviewers for their detailed and thoughtful feedback. We are glad to see that all 4 reviewers recommend acceptance, with the reviewers recognizing the ingenuity in scoping the OSLS setup (R1, R2, R3, R4), the novelty of the PULSE framework (R1, R2, R3), and the significance of theoretical and empirical results (R1, R2, R3, R4).

1. Inspired by the feedback from the reviewers, we have significantly improved the exposition of the paper, specifically Section 6 of the paper. We have also updated the description of Algorithms 2, 3, and 4 in Appendix C of the paper.

2. As per suggestions from the reviewers, we have also included results on two more datasets:

  - **Age prediction task. As per the suggestion of Reviewer dUdt,  we have now included experiments on an age-prediction task (UTK-Face) in Appendix F.9 in Table 6 to simulate problems where strong positivity might not hold.** We observe that the prevalence of the novel class as estimated with our PULSE framework is significantly closer to the true estimate. Additionally target classification performance of OSLS is similar to that of $k$PU both of which significantly improve over domain discriminator and source only baselines.



| Method  |   Acc | MPE|
|--------------|-------|------|
| Source Only  |   50.1| 0.11 |
| Domain Disc. |   52.4 | 0.08 |
| kPU          |   **56.7**  | 0.11 |
| PULSE (Ours) | **56.8**  | **0.01** |

  - **FMoW-Wilds. We include this experiment to present preliminary OSLS results on a real-world problem.** For the source domain, we use the training data and for the unlabeled target domain, we use the (i) ID val dataset and (ii) OOD val dataset. In OOD setting, by default going from train data to OOD val presents a shift in the label distribution. To simulate the open set problem, we restricted source classes to 46 and used the data from the rest of the 16 classes as data from a novel class. Note that here along with changing prevalence of different classes from source to target, in OOD case, p(x|y) also varies slightly due to source and target data collected over different (non-overlapping) years. Note for OOD case, we also compare with an oracle PULSE method where we have access to the true target marginal. The table below shows preliminary results:

| | FMoW-Wilds (ID)  | FMoW-Wilds (OOD)
| ---- | ---------- | ---------- |
| Method | Acc &nbsp; &nbsp; &nbsp; &nbsp; &nbsp; MPE | Acc  &nbsp; &nbsp; &nbsp; &nbsp; &nbsp; MPE
| Source Only | 35.4 &nbsp; &nbsp; &nbsp; &nbsp; &nbsp; - | 35.7 &nbsp; &nbsp; &nbsp; &nbsp; &nbsp; -
| Domain Disc. | 45.2 &nbsp; &nbsp; &nbsp; &nbsp; &nbsp; 0.22 | 31.9 &nbsp; &nbsp; &nbsp; &nbsp; &nbsp; 0.6
| kPU | 55.7  &nbsp; &nbsp; &nbsp; &nbsp; &nbsp; 0.44 | 34.1 &nbsp; &nbsp; &nbsp; &nbsp; &nbsp; 0.51
| PULSE | **62.4**  &nbsp; &nbsp; &nbsp; &nbsp; &nbsp; 0.15| 33.9 &nbsp; &nbsp; &nbsp; &nbsp; &nbsp; 0.52
| PULSE (Oracle target marginal) | -  &nbsp; &nbsp; &nbsp; &nbsp; &nbsp; - | 46.5 &nbsp; &nbsp; &nbsp; &nbsp; &nbsp; -


 We respond further to each reviewer’s individual concerns in the respective threads.

---

### Meta-Review · Area_Chair_eEaZ · 2022-08-26

**Recommendation:** Accept
**Confidence:** Less certain

**Metareview:**

The paper addresses an interesting domain adataption question and proposes an novel and elegant
solution supported with relevant theory. Although some issues have been raised, all reviewers agree
that the paper worth be published, and we expect the authors to take into account the comments
of the reviewers (eg discussing limitations of PULSE, checking positivity conditions...)

**Award:**

No

---

### Decision · Program_Chairs · 2022-09-14

Accept